# OpenFake: An Open Dataset and Platform Toward Real-World Deepfake Detection

## Abstract

Deepfakes, synthetic media created using advanced AI techniques, pose a growing threat to information integrity, particularly in politically sensitive contexts. This challenge is amplified by the increasing realism of modern generative models, which our human perception study confirms are often indistinguishable from real images. Yet, existing deepfake detection benchmarks rely on outdated generators or narrowly scoped datasets (e.g., single-face imagery), limiting their utility for real-world detection. To address these gaps, we present OpenFake, a large politically grounded dataset specifically crafted for benchmarking against modern generative models with high realism, and designed to remain extensible through an innovative crowdsourced adversarial platform that continually integrates new hard examples. OpenFake comprises nearly four million total images: three million real images paired with descriptive captions and almost one million synthetic counterparts from state-of-the-art proprietary and open-source models. Detectors trained on OpenFake achieve near-perfect in-distribution performance, strong generalization to unseen generators, and high accuracy on a curated in-the-wild social media test set, significantly outperforming models trained on existing datasets. Overall, our results offer encouraging evidence that detectors trained with high-quality data can generalize to real-world social-media distributions.

## 1 Introduction

Deepfakes, realistic synthetic media generated by AI, have emerged as a serious threat to the information ecosystem (Canadian Security Intelligence Service, 2023; Bengio et al., 2025c). By enabling anyone to fabricate audio-visual content of real people, deepfakes can spread false information at an unprecedented scale, eroding trust across various platforms, from social media and online content to traditional media outlets. High-profile cases (e.g., forged speeches or imagery of public figures) and the prevalence of non-consensual intimate imagery underscore the potential for harm to political stability, reputation, and public safety Marchal et al. (2024). Scholars have warned of an "infopocalypse" where constant exposure to fake media breeds cynicism or paranoia Schick (2020). Detecting deepfakes reliably is therefore critical to mitigate the spread of misinformation and disinformation[1], and to restore trust in digital media. The rapid advancement of AI-generated image technologies has reached a point where distinguishing between real and synthetic images has become increasingly challenging for humans. Studies have shown that humans underperform in identifying AI-generated images, highlighting the sophistication of these generative models (Diel et al., 2024).

The political sphere is particularly vulnerable to the risks posed by deepfakes, which can be weaponized to manipulate public opinion and undermine democratic processes (Bengio et al., 2025c;b; Karen Hao, 2019). Synthetic media have already been exploited for scams, blackmail, and targeted reputation sabotage, while the fabrication of fake historical artifacts, manipulated medical images, and staged events introduces new avenues for the spread of misinformation and societal harm (Ferrara, 2024; Bengio et al., 2025c). By flooding social and traditional media with convincing falsehoods, deepfakes erode public trust in news and create confusion about what is real, particularly during sensitive periods like elections (IVADO and CEIMIA, 2025). Such disruptions threaten not only indi-

---

[1] We adopt the term *misinformation* throughout this paper to refer broadly to harmful or misleading content. Technically, *misinformation* denotes false information shared without intent to deceive, while *disinformation* refers to deliberately deceptive content. Our usage includes both, given the difficulty of inferring intent.

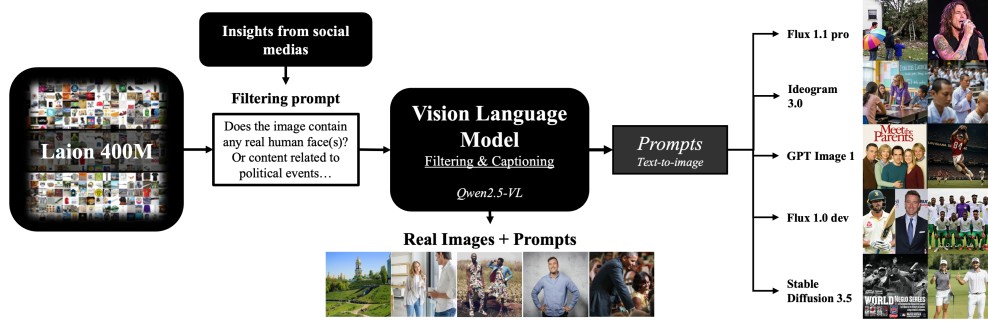

Figure 1: We begin by scraping politically relevant images from social media (e.g., X, Reddit, Bluesky), filtered by election-related hashtags. Manual investigation of these social media images helps us to design a prompt for filtering politically relevant images. A vision-language model (e.g., Qwen2.5-VL) extracts thematic captions or prompts from real images from LAION. These prompts serve dual purposes: (1) forming a large bank of real image–prompt pairs, and (2) seeding generation across a range of synthetic image models (e.g., SDv3.5, Flux, Ideogram, GPT Image 1).

vidual reputations and public safety, but also the legitimacy of democratic institutions and processes, with scholars warning that advances in generative AI could empower malicious actors to influence political outcomes and destabilize societies (Bengio et al., 2025a; Hameleers et al., 2024; Vaccari & Chadwick, 2020).

Despite significant progress in deepfake detection research (LI et al., 2024; Shao et al., 2025; Cozzolino et al., 2024), current datasets suffer from major limitations that restrict their effectiveness in modern, real-world scenarios (Pal et al., 2024; Chen & Zou, 2023). As shown in Table 1, most established benchmarks rely on outdated generation methods; GAN-based face-swapping tools such as DeepFaceLab (Liu et al., 2023) and Face2Face (Thies et al., 2016). These datasets, while valuable for early detection efforts, are increasingly unrepresentative of the latest synthetic media, particularly high-fidelity diffusion and transformer-based models. Moreover, they overwhelmingly focus on single-face portraits, providing little to no real-world grounded images and neglecting the broader spectrum of image-based misinformation that floods political and social media discourse: crowd scenes, protests, disaster images, manipulated signage, or synthetic screenshots.

To address these gaps, we introduce OPENFAKE[2] , a politically grounded dataset for general deepfake detection. OPENFAKE pairs large-scale real image corpora with state-of-the-art synthetic images and is designed to remain extensible through OPENFAKE ARENA, a crowdsourced adversarial platform that continually contributes hard, validated examples via a CLIP-based prompt-consistency gate and scoring against a live detector. This yields a self-improving benchmark that tracks the evolution of modern generators.

By training a modest SwinV2-Small (Liu et al., 2021b) detector on OPENFAKE we produce near perfect in-distribution results on the held-out test sets (unseen images from the seen generators and their variants), as well as strong performance on images from unseen generators (see Table 3), with overall F1 score of 0.99 compared to the 0.88 from our strongest baseline (same model trained on GENIMAGE). More interestingly, the model trained on OPENFAKE achieves a strong performance on a curated in-the-wild social-media test set, with a F1 score of 0.86 compared to 0.08 (GENIMAGE), and 0.26 (SEMI-TRUTHS) (see Table 4). This finding stands in contrast to the prevailing pessimism around automatic deepfake detection, which is deemed as both futile and intractable, largely due to the increasing realism of generative models (Helmus, 2022; Engler, 2019; Kusnezov et al., 2023; Lazerowitz, 2024; Cao, 2019; Babaei et al., 2025). Indeed, some experts argue that distinguishing real from synthetic images will soon become impossible, which has shifted attention to watermarking (Wen et al., 2023; Min et al., 2024; Liu et al., 2025; Saberi et al., 2024). However, watermarking depends on developer cooperation, consistent deployment across proprietary systems, and robustness to post-processing, and therefore cannot replace open-world detection. OPENFAKE paves the way for automatic detection, demonstrating that highly realistic deepfake, which evade detection by human eyes, can be detected with high accuracy. In summary, our contributions are:

• Providing OPENFAKE, an *easily accessible* dataset, which has

---

[2]https://huggingface.co/datasets/Anonymous460/OpenFake

- **Rich Scope**: A large, politically relevant dataset of 3 million real images paired with extracted prompts, curated for misinformation risk and designed by studying real-world social media.
- **High Realism**: A diverse, high-quality synthetic image set spanning 963k images, generated from state-of-the-art open-source and proprietary models.
- **Extendable**: A scalable crowdsourcing framework (OPENFAKE ARENA) for adversarial image generation, enabling continual community-driven benchmarking and available prompts for generating new synthetic images.

- An experimental study which shows *the weak performance of detectors trained on the currently available datasets* on detecting realistic deepfakes, with F1 ranging between 0.0 to 0.88. Our model trained on OPENFAKE significantly outperform these baselines with a F1 of 0.99.

- A human perception study showing that *images from modern proprietary generators can be imperceptible to humans,* with accuracy in some cases dropping to near random chance (e.g., with Imagen 3). Our model trained on OPENFAKE, however, achieves a near perfect performance.

- A real-world feasibility study which report *a strong performance of* **OPENFAKE** *on detecting real and fake images actually circulated on social media*, based on a carefully curated in-the-wild test set. OPENFAKE achieves a F1 score of 0.86, significantly higher that when relying on two strong contenders: 0.08 (GENIMAGE), and 0.26 (SEMI-TRUTHS).

Together, OPENFAKE and OPENFAKE ARENA form a robust and adaptive foundation for studying deepfakes in politically sensitive contexts, providing researchers and practitioners with the publicly available tools necessary to characterize emerging synthetic threats.

## 2 RELATED WORK

**Synthetic image datasets.** Despite the proliferation of generative models, existing deepfake datasets remain limited in realism, diversity, and accessibility. Early benchmarks such as FaceForensics++ (Rössler et al., 2019), Celeb-DF (Li et al., 2020), and DFDC (Dolhansky et al., 2020) rely on GAN-based face-swapping techniques and focus almost exclusively on single-person portrait videos. Even newer datasets such as ForgeryNet (He et al., 2021), OpenForensics (Le et al., 2021), and FFIW (Zhou et al., 2021) continue to emphasize face-centric detection, with limited variation in image content or generation method (Cheng et al., 2024; Chen et al., 2024c; Yan et al., 2024). More recent image datasets have started to incorporate diffusion-based generators (e.g., Stable Diffusion, DALL·E 2, Midjourney), as seen in Fake2M (Lu et al., 2023), DiffusionForensics (Wang et al., 2023), and GenImage (Zhu et al., 2023). However, these datasets still fall short in several ways. First, most rely on open-source models like SDv1.5 or SDv2.1 (Rombach et al., 2022), which, while important, do not match the visual fidelity of cutting-edge proprietary models such as Imagen 3 (Baldridge et al., 2024) or GPT Image 1 (OpenAI, 2025). As a result, they fail to represent the modern threat landscape posed by the most deceptive fakes. Second, many datasets lack real-world grounding. Image prompts are frequently abstract, artistic, or class-based (e.g., GenImage uses classes from ImageNet-1k (Deng et al., 2009)), failing to capture the multimodal misinformation strategies actually deployed online. Third, these datasets are static and infrequently updated, meaning they quickly become outdated as generation tools evolve. Fourth, prompts used for image generation are often withheld, making it difficult for others to reproduce, regenerate, or expand these datasets with future models. In contrast, we release a large bank of extracted prompts along with the images, which enables researchers to extend the dataset. Finally, accessibility remains a persistent issue. Many datasets require downloading large zip archives via Google Drive or web links, making them difficult to integrate into new pipelines. In contrast, OPENFAKE is fully hosted on the HuggingFace Hub in streaming-friendly Parquet format, enabling scalable access and evaluation, which should help the community develop new detection tools. Table 1 highlights these differences in model coverage, dataset scope, prompt extensibility, and access modality.

**Deepfake detection methods.** Early detection approaches relied on convolutional neural networks trained on known forgery artifacts, such as blending boundaries or low-level inconsistencies in the images (Afchar et al., 2018; Rössler et al., 2019; V & Joy, 2023). While effective in-domain, these models struggle to generalize across generation techniques (Ojha et al., 2023). As diffusion and transformer-based models reduce such artifacts, recent work has focused on semantic-level signals and frequency-domain cues (Durall et al., 2020; Liu et al., 2021a; Frank et al., 2020; Qian et al.,

| Dataset | Year | Fakes | Reals | Extends | Content Scope | Realism | Access | Methods | Most recent model |
|---|---|---|---|---|---|---|---|---|---|
| FaceForensics++ (Rössler et al., 2019) | 2019 | 5K | 1K | ✗ | Narrow | Low | Hard | 4 | Face2Face (2016) |
| Celeb-DF (Li et al., 2020) | 2020 | 5K+ | 590 | ✗ | Narrow | Low | Hard | 1 | DeepFaceLab (2020) |
| DFDC (Dolhansky et al., 2020) | 2020 | 100K+ | 20K+ | ✗ | Narrow | Low | Hard | 8 | DeepFaceLab (2020) |
| ForgeryNet (He et al., 2021) | 2021 | 1.5M | 1.5M | ✗ | Narrow | Low | Hard | 15 | DeepFaceLab (2020) |
| FFIW (Zhou et al., 2021) | 2021 | 10K | 10K | ✗ | Narrow | Low | Hard | 3 | DeepFaceLab (2020) |
| OpenForensics (Le et al., 2021) | 2021 | 100K | 100K | ✗ | Narrow | Low | Hard | 3 | GAN (2020) |
| DeepFakeFace (Song et al., 2023) | 2023 | 90K | 30K | ✗ | Narrow | Medium | Hard | 3 | SD v1.5 (2022) |
| Fake2M (Lu et al., 2023) | 2023 | **2M** | 0 | ✗ | Moderate | Medium | **Easy** | 3 | SD v1.5 (2022) |
| DiffusionForensics (Wang et al., 2023) | 2023 | 570K | 140K | ✗ | Moderate | Medium | Hard | 8 | iDDPM (2021) |
| DMDetection (Corvi et al., 2023) | 2023 | 200K | 200K | ✗ | Moderate | Medium | Hard | 3 | DALL·E 2 (2022) |
| GenImage (Zhu et al., 2023) | 2023 | 1.3M | 1.33M | ✗ | Moderate | Good | Hard | 5 | Midjourney 5 (2023) |
| TWIGMA (Chen & Zou, 2023) | 2023 | 800K | 0 | ✗ | Rich | Medium | Unavailable | – | – |
| DiffusionDeepfake (Bhattacharyya et al., 2024) | 2024 | 100K | 94K | ✗ | Narrow | Good | Hard | 2 | Midjourney (2024) |
| DF40 (Yan et al., 2024) | 2024 | 1M+ | 1.5K | ✗ | Narrow | Medium | Hard | **40** | PixArt-α (2024) |
| DiffusionFace (Chen et al., 2024c) | 2024 | 600K | 30K | ✗ | Narrow | Good | Hard | 11 | SD v2.1 (2022) |
| DiFF (Cheng et al., 2024) | 2024 | 500K | 23K | ✗ | Narrow | Good | Hard | 13 | Midjourney 5 (2023) |
| Semi-Truths (Pal et al., 2024) | 2024 | 1.34M | 26K | ✗ | Moderate | Good | **Easy** | 8 | Stable Diffusion XL (2023) |
| OPENFAKE (Ours) | 2025 | 963K | **3M** | ✓ | **Rich** | **High** | **Easy** | 18 | Imagen 4.0 (**2025**) |

Table 1: **Compared to current public deepfake datasets, OpenFake uniquely combines rich scope, high realism, large real sample count, easy access, and extensibility.** "Fakes" and "Reals" count individual media items (images or videos; units omitted for brevity). **Content Scope**: Narrow (face-focused); Moderate (diverse but limited); Rich (broad, internet-like variety). **Access**: Unavailable; Hard (public but cumbersome); Easy (clean, ready-to-use hosting). **Realism**: qualitative fidelity of synthetic content — Low, Medium, Good, High.[3] **Extendable**: availability of prompts/metadata enabling easier dataset expansion.

2020). CLIP-based detection (Cozzolino et al., 2024; Khan & Dang-Nguyen, 2024) has emerged as a promising direction, leveraging large-scale vision-language pretraining to improve robustness. Other advances include domain-adaptive feature learning (Shao et al., 2025; Jia et al., 2024), zero-shot detectors (Lin et al., 2024), and hybrid approaches that blend local artifact patterns with global semantic reasoning (LI et al., 2024; Zhou et al., 2024; Ma et al., 2025). Despite progress, the rapid pace of generative model development continues to outstrip detection capabilities, motivating adaptive benchmarks like OPENFAKE ARENA to assess robustness in a dynamic, adversarial setting.

## 3 CASE STUDY: REAL-WORLD MISINFORMATION AND HUMAN LIMITS

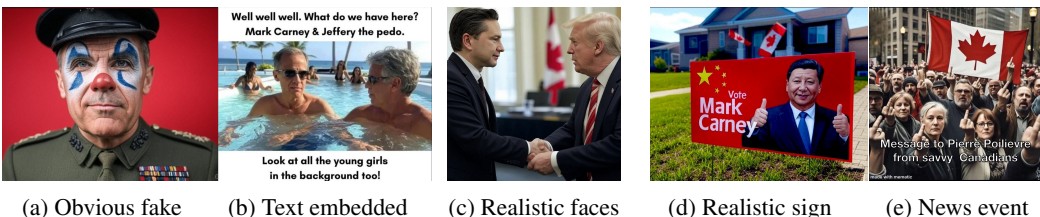

(a) Obvious fake  (b) Text embedded  (c) Realistic faces  (d) Realistic sign  (e) News event

Figure 2: Examples of deepfake images collected from X depicting various types of fabricated scenarios involving Canadian political figures and events.

Social media platforms have become critical channels for political discourse, and consequently, for amplifying deepfake disinformation. This raises a key question: how are deepfakes actually deployed within political conversations? As part of a subsequent study, and to investigate how deepfakes are used in political conversations and to later evaluate our detector in a realistic setting, we collected images from X, Reddit, and Bluesky. The collection spans the period immediately before and during the 2025 Canadian federal election, enabling analysis of synthetic media in a high-stakes, time-sensitive information environment.

We manually examined over 2000 randomly sampled images collected during a 72-hour period to better understand the types of visuals circulating on social media, identifying 163 deepfakes (see

---

[3]The realism score reflects the highest-quality generator included in each dataset. *Low* realism denotes synthetic content with obvious artifacts such as duplicated limbs, distorted structure, or major semantic inconsistencies. *Medium* realism corresponds to mostly coherent images that still exhibit glitches or local distortions. *Good* realism indicates high overall fidelity with few subtle imperfections. *High* realism is reserved for state-of-the-art generators that produce visually consistent, semantically plausible images with minimal detectable artifacts. OPENFAKE contains many of the latest 2025 generators (Flux, Imagen, Ideogram, Midjourney, ect.)

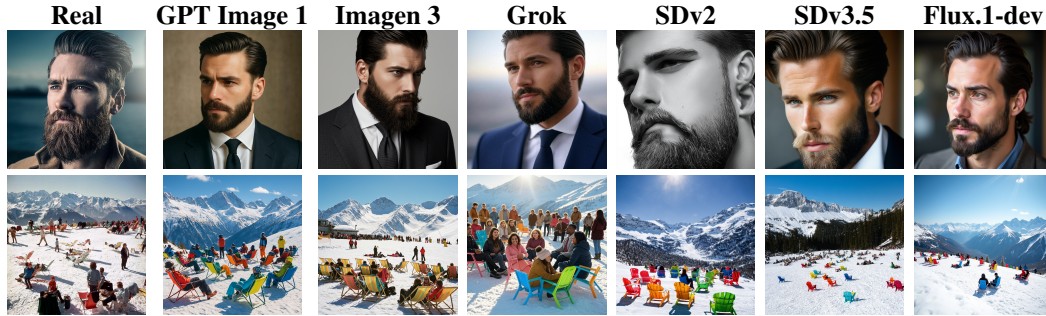

| Real | GPT Image 1 | Imagen 3 | Grok | SDv2 | SDv3.5 | Flux.1-dev |
|------|-------------|----------|------|------|--------|------------|

Figure 3: Examples of deepfakes from each model used in the survey with their respective real image.

Section 6.1 for more details). Many of these prominently featured political figures. Fabricated scenarios involving leading candidates distorted public perception, and even when some deepfakes were clearly artificial (Figure 2a), they reinforced existing biases more effectively than textual misinformation alone (Ecker et al., 2022; Hameleers et al., 2020; Vaccari & Chadwick, 2020). When photorealistic deepfakes aligned with viewers' prior beliefs, the risk was higher, as illustrated in Figure 2c. Beyond portraits, misinformation extended to political symbols, banners, and manipulated depictions of protests or disasters (Figure 2d,e), often paired with misleading text (Figure 2b). These findings inform the construction of our dataset: to support generalizable detection, a deepfake benchmark must move beyond faces to capture the full breadth of misleading visual content. While this dataset serves here to illustrate the variety of real-world deepfakes, in Section 6.1 we also use it as a small in-the-wild evaluation set. Importantly, none of these images or their captions are included in training or generation, ensuring a clean separation between evaluation and benchmark design.

## 3.1 HUMAN PERCEPTION STUDY

To assess the difficulty of detecting deepfakes generated by different models, we conducted a simple human study [4]. A total of 100 participants completed the survey, each viewing a randomized set of 24 images. The set consisted of 12 real photographs and 12 synthetic images—2 from each of the six generation models: GPT Image 1 (OpenAI) (OpenAI, 2025), Imagen 3 (Google) (Baldridge et al., 2024), Grok 2 (xAI), Flux.1.0-dev (Black Forest Labs) (Labs, 2024), Stable Diffusion 2.1 Rombach et al. (2022), and Stable Diffusion 3.5. All synthetic images were generated from the same prompts as their real counterparts, using the text automatically extracted by our pipeline described in Section 4. Each prompt was only shown once to a given participant. This ensured that responses reflected a fair and diverse exposure across the dataset. In total, the survey contained 168 unique images.

| Source | Release | Access | Humans $\kappa$ | Humans $d'$ | Humans $c$ | Humans Accuracy | CLIP-D-10k+ | Corvi2023 | Fusion (CLIP+Corvi) | SwinV2 (GenImage) | SwinV2 (Semi-Truths) | SwinV2 (OPENFAKE) |
|--------|---------|--------|--------|--------|--------|--------|--------|--------|--------|--------|--------|--------|
| Real | — | — | 0.139 | | | 0.718 | 0.479 | 0.000 | 0.062 | 1.000 | 0.500 | **1.000** |
| Imagen 3 | 2024 | Proprietary | 0.074 | 0.553 | 0.301 | 0.490 | 0.458 | 0.708 | 0.667 | 0.625 | 1.000 | 1.000 |
| GPT Image 1 | 2025 | Proprietary | 0.142 | 1.069 | 0.043 | 0.684 | 0.458 | 0.500 | 0.458 | 0.750 | 1.000 | 1.000 |
| Flux.1.0-dev | 2024 | Open | 0.119 | 1.069 | 0.043 | 0.562 | 0.792 | 0.812 | 0.917 | 1.000 | 1.000 | 1.000 |
| SDv3.5 | 2024 | Open | 0.145 | 1.124 | 0.015 | 0.709 | 0.521 | 1.000 | 0.938 | 0.750 | 1.000 | 1.000 |
| SDv2.1 | 2022 | Open | 0.132 | 1.737 | -0.291 | 0.879 | 0.646 | 1.000 | 0.938 | 0.917 | 1.000 | 1.000 |
| Grok 2 | 2024 | Proprietary | 0.214 | 1.453 | -0.149 | 0.811 | 0.771 | 1.000 | 0.938 | 0.208 | 1.000 | 1.000 |
| Overall Metric | | | 0.332 | 1.132 | 0.012 | 0.714 | 0.524 | 0.417 | 0.427 | 0.804 | 0.750 | **1.000** |

Table 2: Survey results showing human true positive rate (TPR) for synthetic images (true negative rate for real images). For human performance, we include Fleiss' $\kappa$ for agreement and signal-detection metrics ($d'$, $c$) for each source. SwinV2 trained on OPENFAKE achieves high accuracy on its in-distribution generators while humans show only moderate consensus.

The results in Table 2 highlight key insights into synthetic image realism and human perception. Imagen 3 from Google achieved the lowest human accuracy (48.5%), equivalent to random guessing. GPT Image 1, OpenAI's recent model, was similarly deceptive, with nearly one-third of its images undetected. In contrast, Stable Diffusion 2.1 (SDv2.1, which is widely used in current benchmarks, had the highest detection rate (87.9%) and was easily flagged by all detectors. This pattern is

---

[4]This human perception study is not demographically representative; participants were primarily drawn from the machine learning community and are likely more familiar with synthetic imagery. As a result, the measured performance likely overestimates what would be observed in the general population.

reinforced by signal-detection metrics: the sensitivity index ($d'$) is highest for SDv2.1 ($d' = 1.737$), confirming that its artifacts remain obvious, whereas sensitivity collapses for Imagen 3 ($d' = 0.553$), showing that participants struggled to differentiate it from real images. Decision-criterion analysis ($c$) further suggests a shift in human strategy: participants adopted a liberal bias ($c < 0$) toward older open-source models, easily labeling them as fake, yet switched to a conservative bias ($c > 0$) for proprietary models, reflecting uncertainty and reduced confidence. Finally, Fleiss' $\kappa$ values remain extremely low across all generators (e.g., $\kappa = 0.074$ for Imagen 3), indicating poor agreement and demonstrating that even when participants correctly identify a deepfake, they do not do so based on consistent or shared cues. These findings suggest that while open-source models remain relatively easy to spot, advanced proprietary models demonstrate exceptional realism and consistency. This underscores the need to include such models in benchmarks, as current deepfake detection datasets trained on older or open-source models fail to capture the new quality standards set by proprietary systems.

However, this perceptual difficulty is not limited to humans. Baseline deepfake detectors also failed to consistently identify these advanced fakes. The CLIP-D-10k+ method (Cozzolino et al., 2024), which fine-tunes a linear classifier on CLIP embeddings, performed close to random on several models (e.g., 45.8% on Imagen 3, and GPT Image 1), and failed to distinguish real images entirely (47.9% TNR). Similarly, the Corvi2023 method (Corvi et al., 2023), which uses curated handcrafted features for detection, fared better on some open-source models (e.g., 100% on SDv3.5), but completely failed on real images (0.0% TNR) and newer proprietary content like GPT Image 1 (50.0%). In contrast, our SwinV2 baseline, trained directly on a curated mix of real and synthetic images from models included in our dataset, achieved 100% overall accuracy and perfect performance on its in-distribution models. While this result demonstrates that deep networks can learn to detect even the most realistic fakes with sufficient supervision, it also highlights the brittleness of existing methods when faced with novel or unseen generative sources. The two additional SwinV2 baselines confirm this pattern: models trained on smaller or differently distributed data (GenImage and Semi-Truths) showed weaker generalization to modern deepfakes, with accuracy dropping to 80.4% and 75.0%, respectively. These results reinforce the importance of training on a large, diverse and higher-quality dataset like OpenFake to ensure robustness against emerging generation techniques.

## 4 DATASET OVERVIEW & COLLECTION

OPENFAKE combines a large repository of real images with a diverse collection of high-quality synthetic counterparts generated by multiple state-of-the-art models. In Appendix A, Table 5 presents in details the key statistics of the dataset. Figure 4 offers a qualitative view of the underlying distribution of some of the images. The substantial overlap between real and synthetic samples in the CLIP feature space highlights their semantic alignment, suggesting that synthetic images effectively mimic the distribution of real-world content. This shows that the prompt generation pipeline is working as intended. Figure 1 presents an overview of the data collection and generation process.

**Real images.** We extract metadata from the LAION-400M dataset (Schuhmann et al., 2021), which we selected due to its broad representation of internet-sourced images—the same domain where visual misinformation typically circulates. Additionally, this dataset was likely included in the training data of the text-to-image models used to generate the synthetic images, which should theoretically make it more difficult for detectors to distinguish between real and fake images. More importantly, these images preserve real-world compression artifacts, which are crucial for training detectors that operate in the wild. While LAION may contain some synthetic images, we expect this contamination to be minimal, as the dataset primarily consists of content from 2014–2021, before the public release of diffusion models in 2022. After scraping LAION, we filter image–caption pairs using a vision–language model (Qwen2.5-VL (Bai et al., 2025)). As described by the prompt used to query the model in Section C, an image is retained if it is identified as depicting either (i) real human faces or (ii) politically salient or newsworthy events. For every retained image, we generate a more detailed caption to use as prompt input for text-to-image models. These $3\,\mathrm{M}$ prompts are also publicly released and form the basis of the prompts shown to users of our crowdsourcing platform OPENFAKE ARENA.

We filtered real images using Qwen2.5-VL, selected for its trade-off between speed and quality. To prevent detection shortcuts, we excluded LAION images smaller than $512{\times}512$ pixels from

the released `train/test` sets, as lower resolutions introduced compression artifacts that made detection artificially easier. Full prompts used for filtering and captioning are provided in Section C, and additional details on generation and compute resources are in Section E.

**Synthetic images.** We generated images from a diverse set of state-of-the-art generators using samples from our prompt bank: Stable Diffusion 1.5/2.1/XL/3.5 (Rombach et al., 2022), Flux 1.0-dev/1.1-Pro/Schnell (Labs, 2024), Midjourney v6/v7 (Midjourney, 2024; 2025), DALL·E 3 (OpenAI, 2023), Imagen 3/4 (Baldridge et al., 2024; Google Cloud, 2025), GPT Image 1 (OpenAI, 2025), Ideogram 3.0 (Ideogram AI, 2025), Grok-2 (xAI, 2024), HiDream-I1 (HiDream-ai, 2025b;a), Recraft v3 (Recraft, 2024), Chroma (Lodestones, 2025), and 10 community variants (Finetuned or LoRA) of Stable Diffusion 1.5/XL and Flux-dev. All images are produced at ∼1 MP resolution with varied aspect ratios (9:16, 16:9, 1:1, 2:3, 3:4, *etc.*), mirroring common social-media formats. Because several proprietary sources impose "non-compete" clauses, those subsets are released under a non-commercial license.

**Splits and accessibility.** We construct the `train/test` split by sampling 1,000 images per generative model (with the exception of out-of-distribution models, which contribute between 200 and 600 images) along with the corresponding number of real images, yielding a test set of roughly 60,000 images (about 3% of the dataset assuming balanced classes). The remaining images are allocated to the training set. To ensure balance, each model is equally represented in the `test` split, and real images are matched accordingly. The rest of the real images and prompts are provided in a CSV file, with the real images accessible through their URLs. As OPENFAKE ARENA expands and more synthetic images are collected, additional real images will be incorporated to preserve parity between real and synthetic domains in the `train/test` splits. All assets are hosted on the HuggingFace Hub. All images have their associated prompts and model name as metadata, which can be used for model attribution.

## 5 CROWDSOURCED ADVERSARIAL PLATFORM

Generative and detection models co-evolve: advances in generation demand stronger detectors, which in turn promote new generation models. To keep benchmarks relevant amid rapid progress, we introduce OPENFAKE ARENA: a crowdsourced platform where users generate synthetic images to fool a live detector. Successful examples are added to the benchmark, enabling sustained evaluation.

OPENFAKE ARENA[5] is designed as a web-based interactive game to encourage wide participation. Each round begins with a prompt sampled from our bank of over 3 million. Users respond by generating a synthetic image using any generative model or editing tool that aligns with the prompt. A CLIP-based similarity gate verifies prompt-image alignment. If the image passes this check, it is evaluated by a detector trained on the OPENFAKE dataset. If the detector misclassifies the synthetic image as real, the user earns a point and the image is added to the benchmark.

The Arena features a real-time leaderboard to gamify the experience and incentivize participation. The detector is periodically retrained with newly collected data, enabling continual improvement. Submitted images are periodically reviewed. This human-in-the-loop setup transforms model drift from a challenge into a feature, allowing the benchmark to evolve organically alongside the state of generative models. Implementation details and screenshots of the arena are in Appendix F.

## 6 BASELINE DETECTOR BENCHMARKS

We evaluate a selection of deepfake detectors on the OPENFAKE dataset, with the goal of assessing how well existing models generalize to modern synthetic media, especially high-quality images from diffusion and transformer-based models.

**Benchmark models.** Our primary detector is **SwinV2-Small** (Liu et al., 2021b), a hierarchical vision transformer that has achieved state-of-the-art results on large-scale classification tasks. We

---

[5]`https://huggingface.co/spaces/Anonymous460/OpenFakeArena`, we provide image examples in Appendix F.

adopt it as the backbone for our supervised detector, trained on OPENFAKE. We also train a **ConvNeXt-V2-base** (Woo et al., 2023), **ViT-base** (Dosovitskiy et al., 2021), and **ResNet-152** (He et al., 2016) for comparison. In addition, we evaluate two SwinV2 variants trained on external datasets (GenImage and Semi-Truths, chosen based on relevance from Table 5), a **ConvNeXt** baseline trained on DRCT taken from (Chen et al., 2024a), and an **EfficientNet-B4** baseline trained on FaceForensics++ from (Yan et al., 2023). For semi-supervised detection, we include the **CLIP-Based Synthetic Image Detector** (Cozzolino et al., 2024), which applies a linear probe over CLIP embeddings with minimal training data. For zero-shot detection, we use **InternVL** (Chen et al., 2024b) directly without finetuning.[6] Finally, we also test the handcrafted detector of Corvi et al. (2023) and a hybrid fusion baseline that averages predictions from CLIP and Corvi2023. Together, these baselines cover a diverse range of architectures, training regimes, and prior benchmarks, allowing us to compare detectors trained on OpenFake against both legacy and contemporary approaches.

| | | OPENFAKE | | | | GenImage | S.-Truths | DRCT | FF++ | CLIP | DMD | InternVL-3 |
| | | SwinV2 | ConvNeXt-V2 | ViT | ResNet-152 | SwinV2 | SwinV2 | ConvNeXt | EffNet-B4 | D-10k+ | Corvi'23 | (zero-shot) |
|---|---|---|---|---|---|---|---|---|---|---|---|---|
| | Real (TNR) | **0.995** | 0.995 | 0.988 | 0.985 | 0.955 | 0.689 | 0.777 | 0.516 | 0.703 | 0.998 | 0.431 |
| In-distribution | SD 1.5 | 1.000 | 1.000 | 0.994 | 0.999 | 0.936 | 1.000 | 0.447 | 0.529 | 0.579 | 0.000 | 0.849 |
| | SD 2.1 | 1.000 | 1.000 | 1.000 | 0.998 | 0.998 | 0.999 | 0.482 | 0.453 | 0.717 | 0.011 | 0.900 |
| | SD XL | 1.000 | 1.000 | 0.999 | 1.000 | 0.956 | 1.000 | 0.426 | 0.507 | 0.438 | 0.001 | 0.814 |
| | SD 3.5 | 1.000 | 1.000 | 0.998 | 0.997 | 0.982 | 1.000 | 0.324 | 0.466 | 0.406 | 0.000 | 0.796 |
| | Flux 1.0 Dev | 1.000 | 1.000 | 0.999 | 0.998 | 0.967 | 0.999 | 0.290 | 0.450 | 0.401 | 0.005 | 0.748 |
| | Flux-1.1-Pro | 1.000 | 1.000 | 0.965 | 0.984 | 0.315 | 0.975 | 0.319 | 0.467 | 0.596 | 0.000 | 0.722 |
| | Flux-1.0-Schnell | 0.999 | 0.998 | 0.985 | 0.996 | 1.000 | 0.998 | 0.289 | 0.476 | 0.503 | 0.000 | 0.803 |
| | Midjourney 6 | 1.000 | 1.000 | 0.995 | 0.998 | 0.090 | 0.949 | 0.166 | 0.486 | 0.100 | 0.000 | 0.884 |
| | Midjourney 7 | 0.994 | 0.995 | 0.945 | 0.969 | 0.952 | 0.997 | 0.264 | 0.484 | 0.404 | 0.001 | 0.961 |
| | DALL·E 3 | 0.995 | 0.997 | 0.990 | 0.993 | 0.238 | 0.927 | 0.461 | 0.543 | 0.394 | 0.000 | 0.983 |
| | GPT Image 1 | 0.998 | 1.000 | 0.994 | 0.993 | 0.772 | 0.983 | 0.402 | 0.442 | 0.384 | 0.005 | 0.932 |
| | Ideogram 3.0 | 1.000 | 1.000 | 0.992 | 0.993 | 0.993 | 1.000 | 0.254 | 0.481 | 0.414 | 0.001 | 0.844 |
| | Imagen 3.0 | 0.999 | 0.996 | 0.980 | 0.978 | 0.962 | 0.998 | 0.237 | 0.461 | 0.286 | 0.005 | 0.784 |
| | Imagen 4.0 | 0.996 | 0.999 | 0.982 | 0.983 | 0.948 | 0.996 | 0.228 | 0.459 | 0.359 | 0.003 | 0.796 |
| | Grok 2 | 1.000 | 1.000 | 0.995 | 0.996 | 0.142 | 0.963 | 0.383 | 0.463 | 0.303 | 0.000 | 0.805 |
| | HiDream-I1 Full | 1.000 | 0.999 | 0.986 | 0.983 | 0.976 | 0.993 | 0.332 | 0.440 | 0.485 | 0.000 | 0.789 |
| | Chroma | 0.992 | 0.992 | 0.941 | 0.952 | 0.980 | 0.995 | 0.451 | 0.435 | 0.298 | 0.003 | 0.726 |
| Out-of-dist. | *Ideogram 2.0* | 0.993 | 0.997 | 0.904 | 0.968 | 0.997 | 1.000 | 0.234 | 0.482 | 0.777 | 0.000 | 0.865 |
| | *Lumina* | 1.000 | 1.000 | 0.995 | 1.000 | 1.000 | 1.000 | 0.494 | 0.355 | 0.720 | 0.028 | 0.983 |
| | *Frames* | 0.968 | 0.968 | 0.908 | 0.948 | 0.816 | 1.000 | 0.368 | 0.392 | 0.920 | 0.000 | 0.912 |
| | *Halfmoon* | 0.995 | 0.958 | 0.847 | 0.874 | 0.953 | 1.000 | 0.263 | 0.353 | 0.632 | 0.000 | 0.832 |
| | *Recraft v2* | 0.972 | 0.965 | 0.986 | 0.965 | 0.699 | 1.000 | 0.379 | 0.443 | 0.248 | 0.004 | 0.929 |
| | *Recraft v3* | 0.701 | 0.694 | 0.773 | 0.775 | 0.288 | 0.997 | 0.364 | 0.497 | 0.430 | 0.002 | 0.912 |
| | Average TPR | 0.988 | 0.981 | 0.963 | 0.971 | 0.823 | 0.992 | 0.354 | 0.475 | 0.443 | 0.003 | 0.827 |
| | Overall F1 | **0.992** | 0.992 | 0.983 | 0.983 | 0.881 | 0.861 | 0.449 | 0.485 | 0.509 | 0.005 | 0.697 |
| | Overall ROC AUC | **1.000** | 0.999 | 0.998 | 0.998 | 0.926 | 0.960 | 0.616 | 0.493 | 0.600 | 0.487 | 0.629 |
| | Overall PR AUC | **1.000** | 0.999 | 0.998 | 0.999 | 0.949 | 0.952 | 0.613 | 0.493 | 0.600 | 0.488 | 0.586 |

Table 3: Performance comparison on OPENFAKE across detectors trained on different datasets. Finetuned (FT) and LoRA variants are grouped under their respective base generators. Generators shown in blue are out-of-distribution for all detectors. SwinV2 trained on OPENFAKE consistently outperforms others on unseen generators, while most alternative detectors exhibit high false positive rates (misclassification of real images). Additional OpenFake-trained backbones (ConvNeXtV2, ViT, ResNet-152) closely track SwinV2, confirming that the gains come from the dataset rather than a particular architecture.

**Results on OPENFAKE.** As shown in Table 3, the SwinV2 trained on OPENFAKE is near-perfect, confirming that modern classifiers are highly reliable when trained on the evaluation distribution. Separately, we evaluate robustness to compression artifacts and, with artifact-matching augmentation, the same model attains an F1 of 0.992 on a fully compressed test set (see Section B.2 for more details). The SwinV2 trained on GENIMAGE is the next strongest and handles many shared open-source families, but degrades on newer proprietary generators (Grok 2, Midjourney 6, Flux-1.1 pro, DALL·E 3), reflecting a distribution gap in the fake images. The SwinV2 trained on SEMI-TRUTHS achieves high TPRs yet misclassifies many real images, which is consistent with its training data focused on edits rather than full generation. Our added OpenFake-trained baselines (ConvNeXt-V2, ViT, and ResNet) show that alternative high-capacity backbones achieve almost identical performance to SwinV2, reinforcing that the dataset, rather than the architecture, is responsible for the improvements. Legacy or narrow baselines (ConvNeXt/DRCT, EffNet-B4/FF++) underperform, CLIP-D-10k+ is middling, and Corvi2023 largely predicts "real," while zero-shot InternVL is better than older baselines but still trails supervised models. These legacy baselines are pretrained detectors rather than models we

---

[6]We use the InternVL3-38B;

retrained on OPENFAKE, which further contributes to their weaker performance. This aligns with recent surveys and benchmarks that find most state-of-the-art deepfake detectors rely on general-purpose CNN or Vision Transformer classifiers, or CLIP-style visual encoders with a light detection head, trained on sufficiently rich datasets rather than on highly specialized forensic architectures (Heidari et al., 2024; Gong & Li, 2024; Cozzolino et al., 2024). In our setting, CLIP-D-10k+ and DMD (Corvi2023) play the role of specialized detectors, yet they lag behind OpenFake-trained SwinV2, ConvNeXt-V2, ViT, and ResNet, which supports the view that coverage and realism of the training data are more critical than architectural specialization for robust detection. Overall, robust performance requires training on the correct, broad, and up-to-date image distribution. This pattern is expected: the baseline detectors were trained on different generator sets and, more importantly, on fundamentally different data distributions. This mismatch explains why models trained on GenImage or Semi-Truths can appear stronger on some unseen generators that also fall outside the scope of OPENFAKE.

**Transferability to unseen models.** Using the out-of-distribution generators in Table 3 (blue rows), which were collected from public web sources rather than generated by us, the SwinV2 trained on OPENFAKE shows the strongest transfer while keeping a high true-negative rate on real images. The SwinV2 trained on GENIMAGE is competitive on several open-source families but lags on newer proprietary models, and the SEMI-TRUTHS model attains high TPRs yet mislabels many real images, so its apparent OOD gains are not reliable. Cross-benchmark tests as seen in Table 6 (Appendix B.1) reinforce this: when evaluated on GENIMAGE, the OPENFAKE model substantially outperforms the SEMI-TRUTHS model (Accuracy 0.849 vs. 0.613; F1 0.836 vs. 0.714), and when evaluated on SEMI-TRUTHS it exceeds the GENIMAGE model (Accuracy 0.920 vs. 0.865; F1 0.947 vs. 0.907), despite being out-of-domain in both cases. This suggests that while **dataset coverage is the main driver of transferability**, there is also some degree of cross-generator generalization, likely because different models share subtle artifacts; the broader the training distribution, the more likely a detector can recognize previously unseen generators.

## 6.1 DETECTOR IN THE WILD

Performance on benchmarks often differs from performance in real-world settings. For deepfake detection, evaluation in the wild is particularly challenging. Manually labelling real images is not always straightforward, since, as discussed in Section 3.1, humans struggle to identify high-quality fakes. Real images are easier to validate because their authenticity can often be established through provenance cues such as credits from a reputable source, multiple photos of the same event, consistent backgrounds, or camera metadata. Images without any such evidence can be discarded as uncertain. The risk, however, is that the benchmark becomes trivial, with fakes limited to the easiest cases. With an experienced labeler, the aid of reverse image search, and contextual text (which may explicitly indicate AI generation), more difficult deepfakes can be identified. Using this approach, we constructed a small evaluation set of social media images, described in Section 3, containing 1,057 real images and 163 labeled as deepfakes by us, and compared the performance of our SwinV2 baseline trained on OPENFAKE, GENIMAGE, and SEMI-TRUTHS (Table 4), since these were the only competitive baselines from Table 3.

| Metric | Train OPENFAKE | Train GENIMAGE | Train SEMI-TRUTHS |
|---|---|---|---|
| TNR | 0.976 | 0.998 | 0.220 |
| TPR | 0.865 | 0.043 | 0.908 |
| Accuracy | **0.962** | 0.871 | 0.312 |
| F1 Score | **0.857** | 0.081 | 0.261 |
| ROC–AUC | **0.978** | 0.557 | 0.634 |

Table 4: Generalization of SwinV2 detectors trained on different benchmarks when evaluated on an *in-the-wild* social-media set (1,057 real, 163 fake; see Section 3). Metrics include TNR (real) and TPR (fake). Training on OPENFAKE yields balanced performance, while GENIMAGE and SEMI-TRUTHS show strong class biases.

The detector trained on OPENFAKE shows encouraging results. It produces very few false positives, meaning real images are rarely misclassified as deepfakes, while still identifying 86.5% of the fakes. Although the evaluation set is small and not without limitations—labels were verified, but some of the most difficult cases may have been discarded during curation (out of roughly 2000 candidate images, many were removed, including irrelevant real samples such as screenshots of text or drawings); the results strongly suggest that OPENFAKE offers superior real-world applicability compared to existing datasets. This is a promising outcome for the deepfake detection community. Expanding generator coverage and incorporating image edits, rather than only fully generated images, could further improve performance and move closer to practical, reliable detection systems.

## 7 CONCLUSION

We introduced OPENFAKE, a politically grounded benchmark built from three million real images paired with nearly one million high-quality synthetic counterparts, and extended it with OPENFAKE ARENA, a crowdsourced adversarial platform for continual updates. Our human perception study confirmed that recent proprietary generators often fool users, while detectors trained on older datasets fail against these models. In contrast, detectors trained on OPENFAKE achieved strong in-distribution performance and promising results on a curated in-the-wild set of social-media images, suggesting that reliable detection of deepfakes is attainable outside controlled benchmarks.

While performance on some proprietary or lower-quality sources remains uneven, the path forward is clear: expanding generator coverage and broadening real image diversity (e.g., camera types, capture conditions) to further improve robustness. By combining high-fidelity benchmarking with community-driven adversarial submissions, our framework aims to narrow the gap between generation and detection, equipping researchers and practitioners with tools to confront emerging misinformation threats in real time.

**Ethics, Privacy & Limitations**   The OpenFake dataset, despite its contributions, faces several limitations and ethical challenges. The current in-the-wild test set is small and curated for label certainty, which, while necessary due to the difficulty of verifying real-world fakes, may not represent the full spectrum of sophisticated deepfakes. Thus, it is hard to say with certainty that the model is working on hard, unseen, deepfakes. Ethically, the use of LAION-400M raises concerns around copyright and potentially harmful content. We mitigated these risks through extensive filtering and by restricting its role to a discriminative classification task aimed at reducing the spread of harmful deepfakes. In this context, the anticipated societal benefit of improving deepfake detection outweighs the limited residual risks associated with using this filtered subset. For a detailed discussion, please refer to Appendix D.

## 8 REPRODUCIBILITY STATEMENT

We have made all components necessary for reproducibility available. Section 4 describes in detail how both the real and synthetic data were collected and generated, and the complete dataset is publicly released through a permanent link (anonymous for now). The code used for training and evaluating all baseline detectors (including model weights) is provided alongside the OPENFAKE and *in-the-wild* datasets, ensuring that the reported experiments can be replicated. Appendix E gives further implementation details, including training procedures, hyperparameters, and compute resources. These resources should enable independent researchers to reproduce our results and extend the benchmarks under comparable settings.

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

## A OPENFAKE COMPOSITION AND LICENSING

Table 5 summarizes the dataset at the generator level, listing each base model and its LoRA or finetuned variants with release month, exact image counts, and license category. In total there are **963,342** synthetic images drawn from Stable Diffusion (1.5/2.1/XL/3.5), Flux (1.0 dev, 1.1 Pro, Schnell), Midjourney (6/7), DALL·E 3, Imagen (3/4), GPT Image 1, Grok 2, Ideogram 3.0, HiDream, Chroma, and Recraft v3. The real corpus contains **3M** filtered LAION-400M images. Some proprietary and out-of-distribution generators appear with smaller totals because they were sourced from external collections rather than produced end-to-end.

Licensing and access are made explicit to support downstream compliance. We label sources as *Community*, *Non-commercial*, or *Non-compete* and include these labels in the release metadata. All manifests are hosted on the HuggingFace Hub in streaming-friendly formats (Parquet and CSV) with per-item metadata such as model family, variant, release month, and prompt text.

| Source | Release (YYYY–MM) | # Images | Licence |
|---|---|---|---|
| *Real (LAION-400M, filtered)* | 2021–08 | *3M | – |
| Stable Diffusion 1.5 | 2022–08 | 76,510 | Community |
| Stable Diffusion 1.5 (base) | 2022–08 | 20,000 | Community |
| Dreamshaper (FT) | 2023–07 | 36,510 | Community |
| EpicDream (FT) | 2023–08 | 20,000 | Community |
| Stable Diffusion 2.1 | 2022–12 | 135,487 | Community |
| Stable Diffusion XL | 2023–07 | 186,666 | Community |
| Stable Diffusion XL (base) | 2023–07 | 40,000 | Community |
| Epic Realism (FT) | 2025–06 | 59,770 | Community |
| Touch of realism (LoRA) | 2025–06 | 32,828 | Community |
| RealVisXL-v5 (FT) | 2024–09 | 29,300 | Community |
| Juggernaut (FT) | 2025–05 | 24,768 | Community |
| Flux 1.0 dev | 2024–08 | 144,788 | Non-commercial |
| Flux 1.0 dev (base) | 2024–08 | 106,796 | Non-commercial |
| Mystic (FT) | 2024–10 | 15,608 | Non-commercial |
| MVC5000 (LoRA) | 2025–07 | 16,244 | Non-commercial |
| Amateur Snapshot Photos (LoRA) | 2025–06 | 4,140 | Non-commercial |
| Realism (LoRA) | 2024–08 | 2,000 | Non-commercial |
| DALL·E 3 | 2023–10 | 33,336 | Non-compete |
| Midjourney 6 | 2023–12 | 50,000 | Non-compete |
| Imagen 3.0 | 2024–08 | 4,032 | Non-compete |
| Flux-1.0-Schnell | 2024–08 | 36,084 | Non-commercial |
| Flux-1.1-Pro | 2024–10 | 29,923 | Non-commercial |
| Recraft v3 | 2024–10 | 1,000 | Community |
| Stable Diffusion 3.5 | 2024–10 | 139,114 | Non-compete |
| Grok 2 | 2024–12 | 9,803 | Non-compete |
| Ideogram 3.0 | 2025–03 | 28,495 | Non-compete |
| Midjourney 7 | 2025–04 | 3586 | Non-compete |
| GPT Image 1 | 2025–04 | 41,315 | Non-compete |
| HiDream-I1 Full | 2025–04 | 27,904 | Community |
| Imagen 4.0 | 2025–05 | 10,721 | Non-compete |
| Chroma | 2025–08 | 4,532 | Community |
| **Total synthetic** | – | **963,342** | – |

Table 5: OPENFAKE statistics. Image counts are exact. *While we release the entire 3M real images and prompts, only a balanced subset is fully uploaded to the HuggingFace Hub to match the number of fake images. The remainder can be downloaded via URLs provided in CSV files on the Hub. LoRA variants ("LoRA") and full finetunes ("FT") are listed on separate, smaller rows directly below their base models.

# B MORE RESULTS

## B.1 CROSS-BENCHMARK GENERALIZATION OF SWINV2

**Summary.** Table 6 compares SwinV2 detectors trained on three datasets and evaluated across two external test suites. The OPENFAKE-trained model attains the best out-of-domain balance between TPR and TNR on both GENIMAGE and SEMI-TRUTHS, translating into stronger Accuracy and F1. In-domain results (italicized) saturate, as expected, but are less informative about generalization.

| Test set | Metric | Train OpenFake | Train GenImage | Train Semi-Truths |
|---|---|---|---|---|
| | TPR | 0.771 | *1.000* | 0.965 |
| | TNR | 0.928 | *1.000* | 0.261 |
| GENIMAGE | Accuracy | 0.849 | *1.000* | 0.613 |
| | F1 Score | 0.836 | *1.000* | 0.714 |
| | TPR | 0.909 | 0.830 | *1.000* |
| | TNR | 0.962 | 1.000 | *1.000* |
| SEMI-TRUTHS | Accuracy | 0.920 | 0.865 | *1.000* |
| | F1 Score | 0.947 | 0.907 | *1.000* |

Table 6: Cross-benchmark generalization of SwinV2 detectors. Italicised numbers indicate *in-domain* evaluations, where the model is tested on the same dataset it was trained on. TPR = true-positive rate (synthetic images), TNR = true-negative rate (real images). All values are shown to three decimal places.

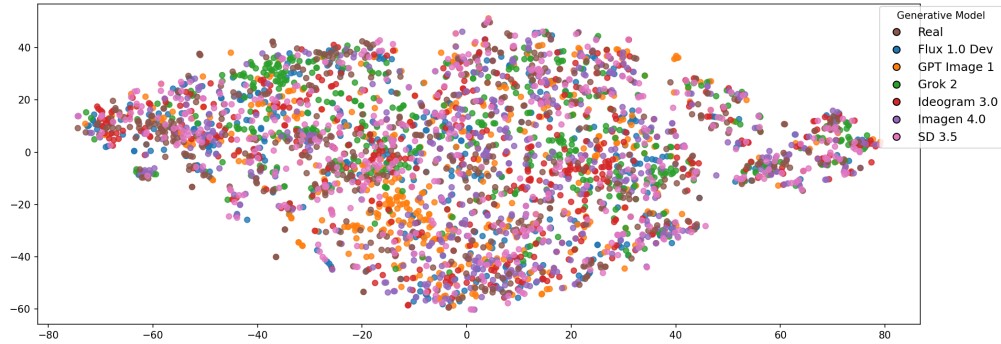

Figure 4: t-SNE visualization of CLIP vision embeddings for 3,500 test images, including both real and synthetic images from a few generative models. Each point corresponds to an individual image, and colours indicate the generative model (or "real" for authentic images).

**Why OPENFAKE transfers better.** We attribute the gains primarily to coverage and recency. OPENFAKE aggregates diverse, up-to-date generators and visual conditions (including compressions and photorealistic prompts), reducing shortcut reliance. By contrast, models trained on older or narrower distributions tend to overfit curation artifacts, which explains the high TPR but poor TNR observed when SEMI-TRUTHS-trained detectors face newer datasets: many real images are misclassified as synthetic.

**Error patterns and operational trade-offs.** The cross-benchmark gaps underscore the importance of calibrating for deployment goals. A detector with inflated TPR but depressed TNR can look strong on superficially balanced metrics yet cause unacceptable false-positive rates in real pipelines. Threshold selection, confidence calibration, and cost-sensitive training are therefore critical when transferring across domains.

## B.2 ROBUSTNESS TO COMPRESSION ARTIFACTS

Real images (sourced from LAION-400M in our dataset) are typically compressed and carry authentic JPEG artifacts and blur. In contrast, synthetic images are high-resolution and minimally compressed. This mismatch creates an obvious signal: detectors may rely on compression differences instead of true semantic features, thus failing on compressed fakes. To assess this vulnerability, we implemented a data augmentation pipeline to degrade synthetic images during training, mimicking the distribution of real images, as done in previous work (Corvi et al., 2023; Wang et al., 2021; Cheng et al., 2024). This includes random resizing, Gaussian blur, JPEG compression, and Gaussian noise. We then evaluated the SwinV2 model trained with these augmentations on a fully compressed test set and got an overall F1 score of 0.992. This demonstrates that the model remains highly accurate even when the compression signal is neutralized.

| | | SwinV2 |
|---|---|---|
| | Real (TNR) | **0.886 (13 152/14 847)** |
| **In-distribution** | SD 1.5 | 0.996 (492/494) |
| | SD 1.5 (Dreamshaper) | 1.000 (488/488) |
| | SD 1.5 (EpicDream) | 1.000 (492/492) |
| | SD 2.1 | 1.000 (508/508) |
| | SDXL | 1.000 (504/504) |
| | SDXL (Epic Realism) | 1.000 (501/501) |
| | SDXL (Juggernaut) | 1.000 (480/480) |
| | SDXL (RealVis v5) | 1.000 (506/506) |
| | SDXL (Touch of Real.) | 1.000 (492/492) |
| | SD 3.5 | 1.000 (498/498) |
| | Flux 1.0 Dev | 1.000 (502/502) |
| | Flux 1.0 Schnell | 0.998 (504/505) |
| | Flux 1.1 Pro | 1.000 (476/476) |
| | Flux Amateur Snap. | 1.000 (502/502) |
| | Flux MVC5000 | 1.000 (491/491) |
| | Flux Realism | 0.998 (466/467) |
| | Midjourney 6 | 1.000 (490/490) |
| | Midjourney 7 | 0.988 (496/502) |
| | DALL·E 3 | 0.994 (477/480) |
| | GPT Image 1 | 0.994 (507/510) |
| | HiDream-I1 Full | 0.998 (496/497) |
| | Imagen 3.0 | 0.998 (490/491) |
| | Imagen 4.0 | 0.989 (519/525) |
| | Grok 2 Image | 0.996 (511/513) |
| | Ideogram 3.0 | 1.000 (504/504) |
| | Aurora-20-1-25 | 1.000 (143/143) |
| | Chroma | 0.980 (486/496) |
| | Mystic | 1.000 (490/490) |
| **Out-of-dist.** | *Ideogram 2.0* | 0.986 (140/142) |
| | *Lumina* | 1.000 (265/265) |
| | *Frames* | 0.901 (127/141) |
| | *Halfmoon* | 0.951 (97/102) |
| | *Recraft v2* | 0.917 (133/145) |
| | *Recraft v3* | 0.503 (254/505) |
| | Average TPR (fakes) | 0.976 |
| | Overall F1 (Argmax) | 0.935 |
| | Overall Acc (Argmax) | 0.932 |
| | Overall ROC AUC | 0.991 |
| | Overall PR AUC | 0.993 |

Table 7: Performance of our OPENFAKE-trained detector on a mixed test built from OPENFAKE and DOCCI ((Onoe et al., 2024)). Real images (*Real* row) consist of 14 847 DOCCI images, while for each generator we subsample a matching number of synthetic images from the OPENFAKE test split. We report per-generator accuracy using the argmax decision rule; generators shown in blue are out-of-distribution for the detector. Despite DOCCI containing recent, high-quality real images, the detector maintains strong performance, indicating that its robustness is not merely due to temporal bias in LAION-style pre-training.

## C  FILTERING AND CAPTIONING OF LAION IMAGES

To curate a relevant subset of real images from LAION-400M, we implemented a two-stage filtering and captioning pipeline using the vision-language model Qwen2.5-VL. This approach allowed us to filter politically salient and emotionally impactful content while preserving real-world visual characteristics (e.g., compression artifacts) crucial for training robust deepfake detectors.

**Filtering prompt.**  The first step used a vision-language reasoning prompt to assess whether each image depicted (i) real human faces, and/or (ii) politically or emotionally significant events. Many original LAION captions are noisy or incomplete, so the model was asked to jointly analyze both image and caption. The prompt was:

```
Analyze the provided image and its caption: "{caption}".
Provide detailed reasoning on the following two points:
1.  Does the image contain any real human face(s)?  Exclude
animations, cartoons, figurines, statues, drawings,
paintings, or video games.
2.  Does the image contain content related to political
events, catastrophes, news events, or anything likely
to have high emotional impact or polarization?  Exclude
animations, cartoons, drawings, paintings, or video games.
Conclude clearly with either "Humans:  yes" or "Humans:
no", and "Catastrophes:  yes" or "Catastrophes:  no".
```

Only images with at least one "yes" label (human or catastrophe) were retained. This strategy allowed us to target both portrait-based and event-based misinformation vectors while filtering out non-photographic and low-impact content.

**Captioning prompt.**  For the selected images, we generated improved prompts to guide synthetic image generation. These prompts describe the image in a style suitable for text-to-image models, incorporating visual format and subject matter. The Qwen2.5-VL prompt used was:

```
Given the image and its caption: "{caption}", generate
a concise prompt in a single sentence that describes the
image and its format (e.g., photograph, poster, screenshot),
including any people present.  Do not mention the caption
directly.
```

These refined prompts were used for synthetic image generation and are also included in the public release to support downstream research and reproducibility.

## D  ETHICS, PRIVACY & LIMITATIONS

**In-the-Wild Evaluation Bias**  Our real-world social media test set consists of a rigorously curated sample of 163 verified deepfakes. To ensure absolute label reliability and provenance, we excluded ambiguous or unverifiable samples during curation. Consequently, while our model demonstrates strong generalization to this set, these results may not fully reflect performance on "hard" examples where ground truth is contested or ambiguous. Since it is inherently impossible to establish verifiable ground truth for most deepfake content circulating freely on social media (due to compression, missing provenance, and re-sharing), rigorous curation for a high-precision holdout set is the only current methodology to provide a reliable, albeit small-scale, assessment of real-world detectability. We intend to expand the diversity and difficulty of this set continuously through community submissions in the OpenFake Arena.

**Data Sourcing and Ethics**  We utilize a subset of LAION-400M as the source for real imagery. We are acutely aware of the community concerns regarding copyright, data quality, and the presence of harmful content in large-scale web crawls. To mitigate these risks, we employed a strict filtering

pipeline (detailed in Appendix C) to remove inappropriate, unsafe, or irrelevant imagery. Crucially, we use this data exclusively for discriminative training (binary classification) rather than for training generative models; we argue that this use case carries significantly lower risk regarding copyright infringement and reproduction of harmful biases than generative pre-training. However, users of OpenFake must still adhere to the licensing terms of the source data.

**Demographic and Geographic Bias**    While our dataset aims to support robust deepfake detection, it inherits limitations from its sources. The real image corpus, derived from the LAION crawl (2014–2021), skews toward Western-centric and pre-pandemic imagery. Proprietary generative models also reflect aesthetic and cultural biases from their training data. These imbalances may affect the generalizability of detection models across diverse global contexts. We document these issues in the HuggingFace Data Card and encourage contributions from underrepresented regions via our Arena pipeline.

The paper includes details of both the human perception study and the Arena crowdsourcing platform. No compensation was offered, as participation was voluntary, and both systems were designed to ensure anonymity and avoid the collection of personal data.

Prompt extraction may introduce semantic noise, and the quality of adversarial data depends on user participation. Our dataset focuses on visual realism, but does not yet capture multimodal or context-based misinformation. Fairness across demographic groups and long-term robustness remain open challenges. We encourage downstream audits and broader evaluation to support responsible deployment.

# E   TRAINING DETAILS AND COMPUTE RESOURCES

## E.1   COMPUTE RESOURCES AND COST

All experiments were conducted on an internal compute cluster or local workstations with moderate storage and GPU availability. Below, we detail the computational resources and costs associated with dataset filtering, image generation, baseline evaluation, and dataset hosting.

**Filtering and analysis.**    The LAION filtering pipeline ran continuously for two weeks on 4 NVIDIA L40S GPUs (48 GB VRAM each). An additional 2 days of compute on the same setup was used for prompt selection and vision–language model evaluation, comparing multiple candidate models and prompt formats.

**Synthetic image generation.**    Images from Stable Diffusion v2.1 and Flux.1.0-dev were generated on 4 L40S GPUs over a span of 4 days per model. Other models generated images for 1 day. Each GPU was fully utilized to maximize throughput.

**Model training and evaluation.**    Training the SwinV2 baseline classifier on the OPENFAKE dataset required approximately 12 hours on a single NVIDIA L40S GPU. Inference for evaluation purposes was negligible in comparison.

**Baseline inference.**    For baseline evaluation:

- InternVL inference over the full test set was performed over 10 hours on a single RTX8000 GPU (48 GB VRAM).
- CLIP and the Corvi2023 baselines were evaluated in approximately 6 hours on the same RTX8000 GPU.

**Proprietary model generation.**    Images generated via proprietary APIs incurred a per-image cost of approximately $0.04 (USD), varying slightly by model and resolution. No GPU compute was required on our end; generation was offloaded entirely to the remote API services.

**Storage and hosting.**    Dataset preprocessing, metadata formatting, and uploads to Hugging Face required only CPU cores but substantial storage capacity. The working set size during dataset preparation exceeded 1TB.

**Total estimated GPU compute:** $\sim$4 GPU-months across L40S and RTX8000 class cards. All compute was performed on institutional resources without incurring cloud costs.

### E.2  SWINV2 FINE-TUNING HYPERPARAMETERS

For our main benchmark detector, we finetune *microsoft/swinv2-small-patch4-window16-256* on the OPENFAKE dataset using the HuggingFace `Trainer` API. All experiments were conducted on a single L40S GPUs.

**Model architecture.**  We use the SwinV2-Small transformer backbone with the classifier head modified to predict two classes: real vs. fake. The model is initialized from ImageNet-1k weights and fine-tuned end-to-end.

**Input resolution.**  Images are resized to $256 \times 256$ using the default SwinV2 image processor.

**Training configuration.**

- **Optimizer:** AdamW
- **Learning rate:** 5e-5
- **Batch size:** 32
- **Epochs:** 5
- **Learning rate scheduler:** Linear with warmup

**Data augmentation.**  During training we use two augmentation streams. A general geometric/photometric stream is applied to *both* real and synthetic images, including random resized crops, color jitter, small rotations, occasional horizontal flips, and mild Gaussian blur. To neutralize compression shortcuts, a light *degradation* stream is applied to *synthetic* images only, including resolution downscaling, blur adjustment, low-level Gaussian noise, and JPEG compression with randomized quality. Transforms are sampled stochastically, and the synthetic-only degradations are calibrated to match statistics of LAION-derived real images. For compressed test-set evaluation, synthetic images are post-processed with the same degradation function to simulate internet-style artifacts; we report accuracy, precision, recall, F1, and ROC AUC.

### E.3  GENERATION PARAMETERS FOR OPEN-SOURCE MODELS

We document here the generation settings used to produce synthetic images from open-source models within the OPENFAKE dataset. This ensures reproducibility and clarity on the diversity of generated outputs.

We used `stabilityai/stable-diffusion-3.5-large` to generate synthetic images and `black-forest-labs/Flux.1.0-dev` using the same bank of prompts. Both models were run in `bfloat16` precision using their official pipelines—`StableDiffusion3Pipeline` and `FluxPipeline`, respectively—and deployed across multiple GPUs with prompt sharding and batched inference for scalability. We used the official HugginFace weights for the other models via the `Diffusers` Python library.

For all models, the following generation settings were generally applied (there could be slight modifications based on the recommended parameters for each model):

- **Resolution**: Randomly sampled from a predefined set of social-media-style sizes: `[(1024, 1024), (1024, 512), (512, 1024), (1024, 768), (768, 1024), (1152, 768), (768, 1152)]`
- **Guidance scale**: Uniformly sampled between 1.5 and 7
- **Inference steps**: [10, 40]
- **Scheduler**: Default

These configurations were chosen to maximize diversity and photorealism, while reflecting the resolution and stylistic variability typical of online content.

# F   OPENFAKE ARENA

We host the Arena as a Gradio app on Hugging Face Spaces, leveraging their compute resources. A pretrained CLIP model acts as a prompt-matching gate to ensure image relevance, and successful submissions that fool the detector are stored in a connected Hugging Face dataset. The detector is a SwinV2 model trained on the OPENFAKE dataset and periodically updated to reflect new data. We also log metadata such as the generative model used and the user ID to support leaderboard tracking. Prompts are designed to be specific and difficult to spoof, and additional safeguards are in place to prevent misuse. Upon acceptance, we plan to promote the Arena through social media and at the conference to encourage broader participation. Figures 5, 7, and 6 show the Arena interface and leaderboard, along with examples of successful and failed submissions.

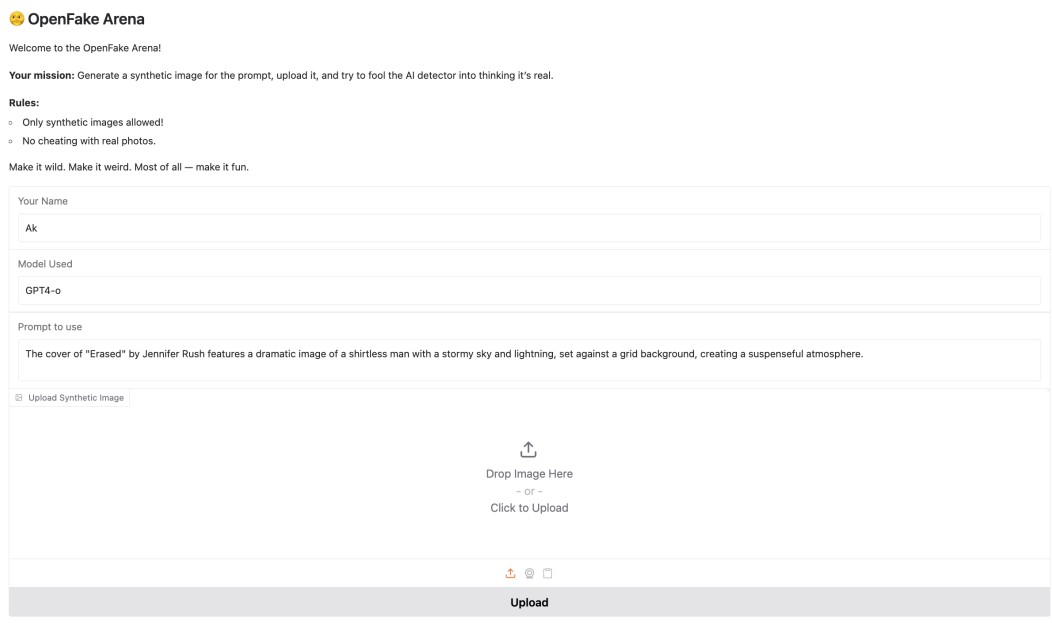

Figure 5: OPENFAKE ARENA interface. Users are presented with a prompt and asked to generate an image that can fool the detector.

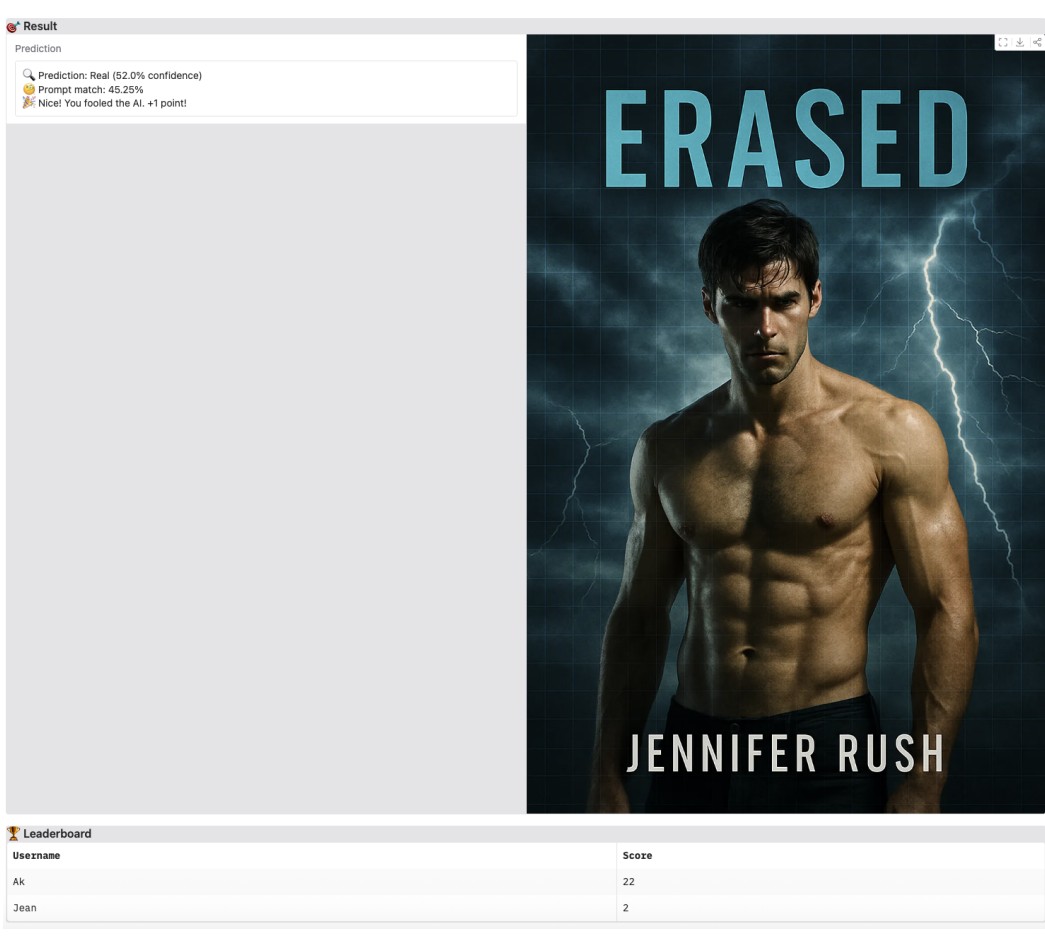

Figure 6: Example of a successful submission. The image aligns with the prompt "The cover of "Erased" by Jennifer Rush features a dramatic image of a shirtless man with a stormy sky and lightning, set against a grid background, creating a suspenseful atmosphere". It is incorrectly classified as real by the detector.

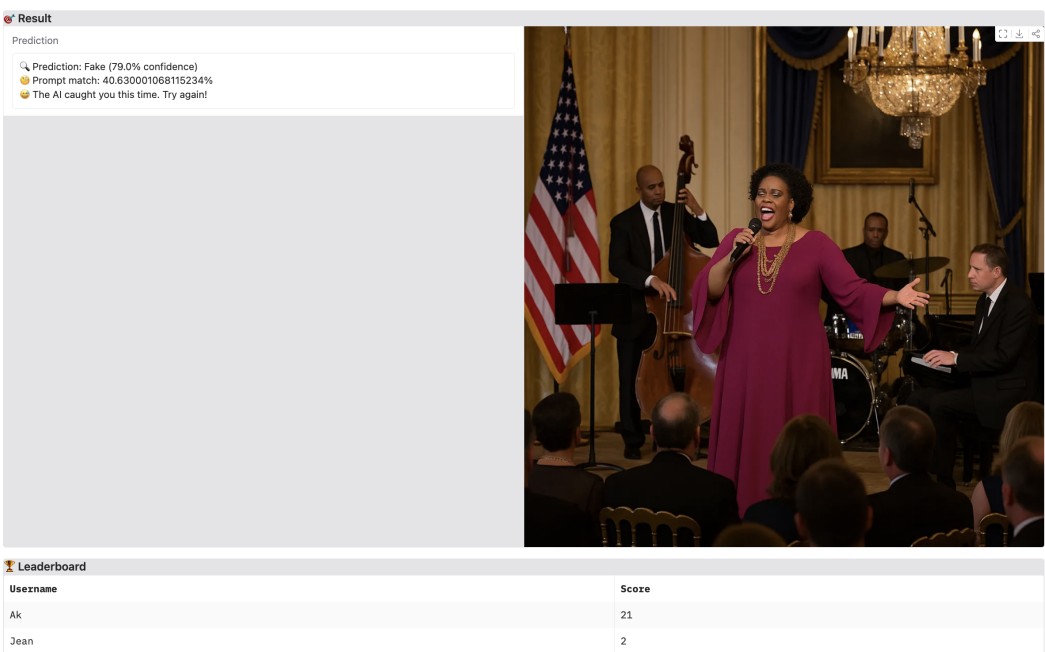

Figure 7: Example of an unsuccessful submission. The image fails to fool the detector and is correctly classified as synthetic. The prompt used was "A photograph captures Dianne Reeves performing on stage in the East Room of the White House during the National Governors Association Dinner on February 26, 2012, with an audience seated in the foreground."

## G SYNTHETIC IMAGE EXAMPLES FROM OPENFAKE

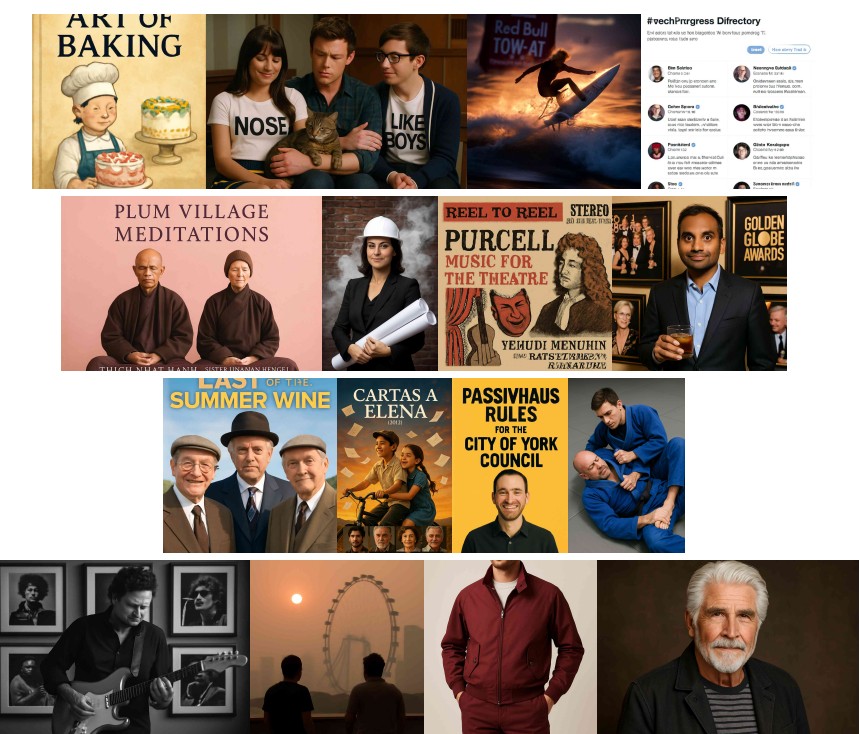

Figure 8: Sample images from OPENFAKE generated by **GPT Image 1**.

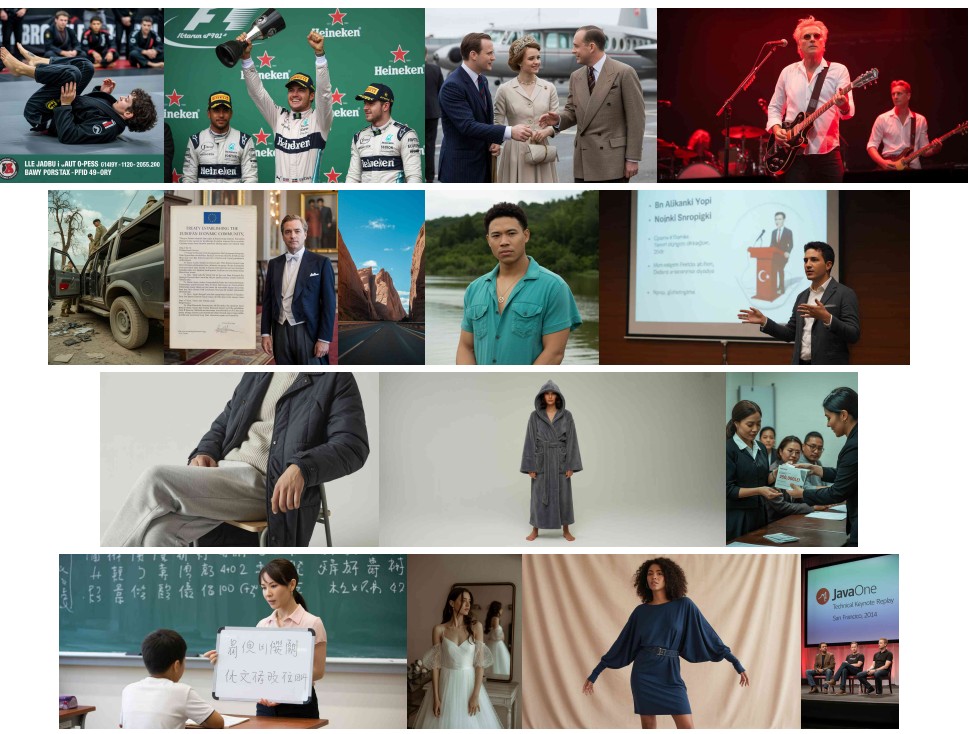

Figure 9: Sample images from OPENFAKE generated by **Ideogram 3.0**.

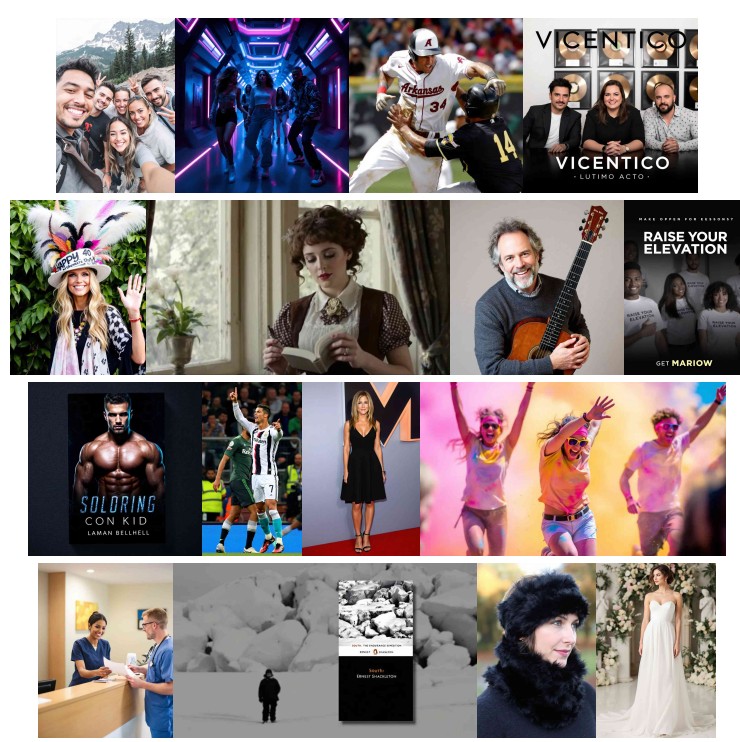

Figure 10: Sample images from OPENFAKE generated by **Flux-1.1-Pro**.

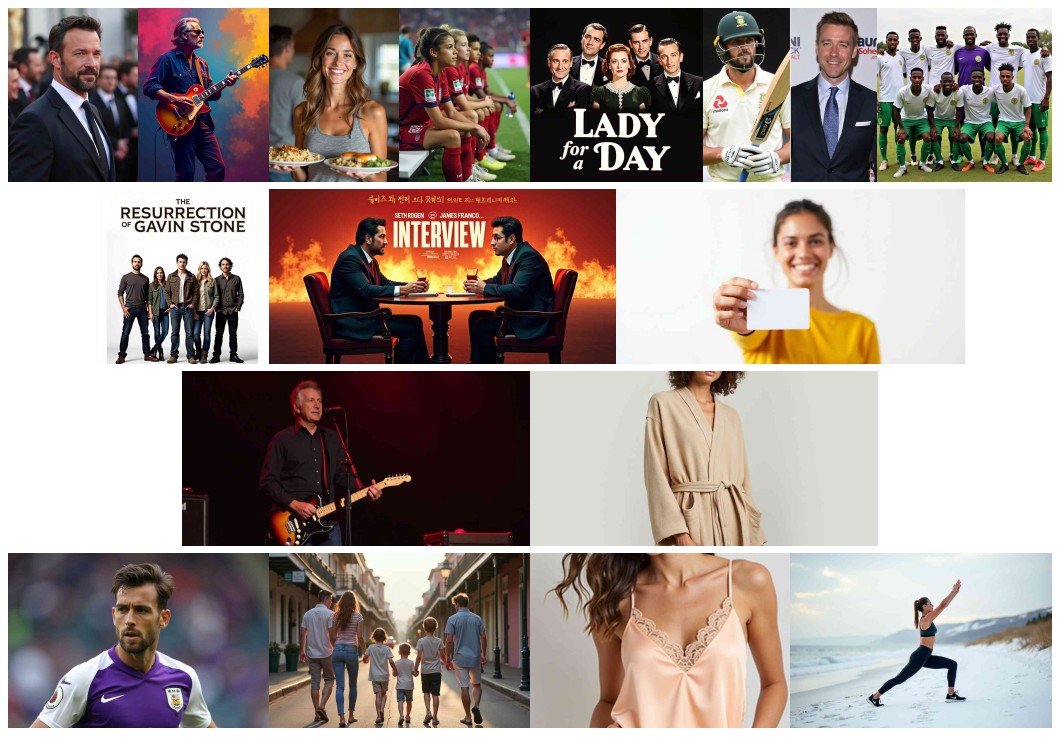

Figure 11: Sample images from OPENFAKE generated by **Flux.1-Dev**.

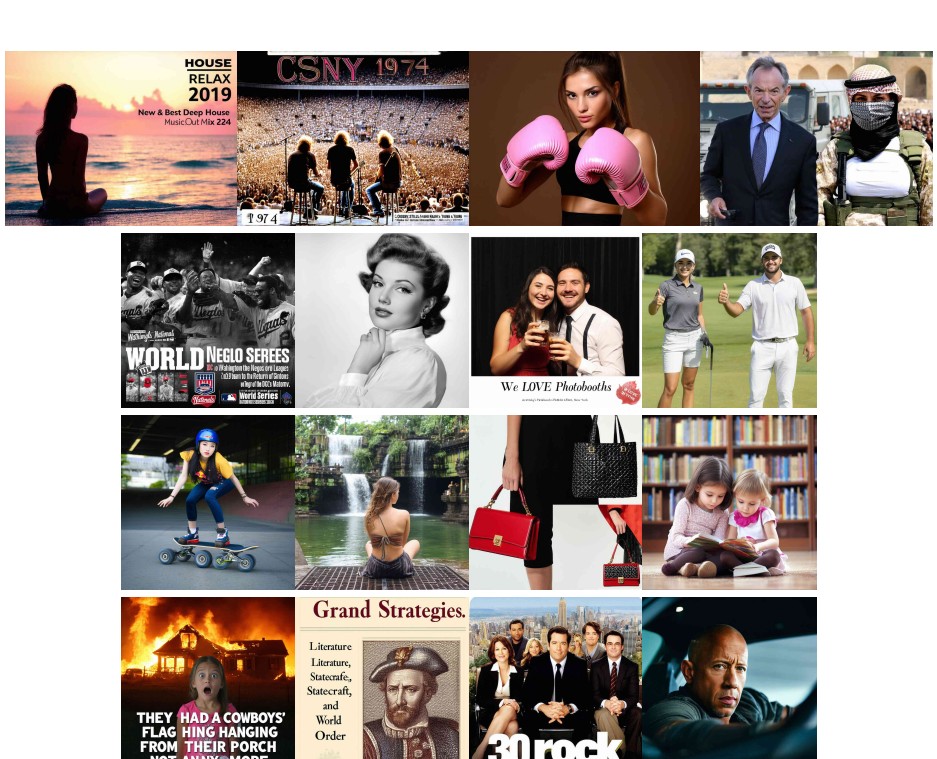

Figure 12: Sample images from OPENFAKE generated by **Stable Diffusion 3.5**.

