# OpenReview forum: "OpenFake: An Open Dataset and Platform Toward Real-World Deepfake Detection"
_ICLR.cc/2026/Conference — Submitted to ICLR 2026_

### Official Review · Reviewer_jNEy · 2025-10-31

**Soundness:** 3
**Presentation:** 3
**Contribution:** 2
**Rating:** 4
**Confidence:** 5

**Summary:**

This paper introduces a new dataset named OpenFake, which collects real images from the LAION-400M dataset using predefined filter prompts and generates fake images with 18 modern generative models, including Stable Diffusion, Midjourney, and Imagen, based on a curated prompt bank. Compared with existing datasets that are often constrained to a limited domain, such as single-face imagery, OpenFake covers a much wider range of scenarios with richer contextual diversity. The dataset also incorporates the latest generative models released in 2025.

In addition, the paper presents a web-based interactive game designed to encourage community participation. Users can create prompts using different generators to produce fake images capable of evading current deepfake detection systems, thereby continuously expanding the dataset’s scale and difficulty. Several experiments are conducted on existing benchmarks to demonstrate the effectiveness of the proposed dataset.

From my perspective, developing new, diverse, and large-scale datasets remains crucial for advancing this field. Hence, OpenFake has the potential to make a valuable contribution and inspire future research.

However, from a research standpoint, this work still faces several fundamental issues that should be carefully addressed.

**Strengths:**

As mentioned in the summary, creating new deepfake datasets that better align with real-world scenarios is always valuable and important, since data itself remains one of the most critical factors driving progress in this field. Moreover, the paper is well written, making it easy to read and follow.

**Weaknesses:**

1. Subjective statements.
   Some claims in the paper appear subjective. For example, in Table 1, the authors define three levels of realism: Low, Medium, Good, and High. However, this categorization lacks objective validation. How are these levels defined, and on what basis is the newly proposed dataset considered “High”? A clear quantitative or perceptual justification is necessary.

2. Unfair experimental setup.
   The experimental design raises fairness concerns. In Table 3, the OpenFake dataset includes all the generators listed in the first column, while other datasets do not. As a result, OpenFake’s in-domain evaluation naturally outperforms others, which is unsurprising. Moreover, the compared datasets listed in the first row are not representative; important benchmarks such as Celeb-DF, DFDC, OpenForensics, and Celeb-DF++ are missing. Additionally, different deepfake detectors are used across datasets (e.g., SwinV2-small is applied to OpenFake, GenImage, and S.-Truth, while less powerful models such as EfficientNet-B4 and ConvNeXt are used for others), which makes the comparison inconsistent and potentially biased.

3. Lack of efficacy demonstration.
   To convincingly demonstrate the effectiveness, it is essential to evaluate state-of-the-art deepfake detectors on OpenFake and examine whether their performance degrades. However, the paper only reports results using SwinV2-Small, without comparisons to other detectors. From my understanding, SwinV2-Small is not a specialized deepfake detector, yet it performs well on OpenFake, which may instead suggest that the dataset is not sufficiently challenging, contradicting the stated motivation of this paper.

4. Questionable definition of “expandable.”
   The concept of building an interactive platform for dataset expansion is indeed interesting and valuable. However, the set of generators is fixed, with no indication of future modification or updates. Changing only the prompts may diversify image content but does not alter the intrinsic generator fingerprint. Consequently, the underlying data distribution remains unchanged, limiting true diversity and questioning the appropriateness of calling the dataset “expandable.”

**Questions:**

NA

---

> ### Author Response · Authors · 2025-11-22
> **Rebuttal**
>
> We sincerely thank the reviewer for the constructive and thoughtful feedback. We appreciate the recognition of the paper’s strengths, particularly the importance of building diverse, large-scale, and realistic deepfake datasets, and we address all concerns in detail below. We have incorporated the suggested clarifications into the revised manuscript.
>
> **[W1]** We agree that “realism” in generative imagery lacks a universally accepted definition. The field currently has no standardized perceptual scale, which we explicitly acknowledge as an open challenge. Our categorization follows common practice in prior deepfake-detection literature, where realism levels are used descriptively to compare capabilities of generative models across years.
>
> To address the reviewer’s concern, we have added a clear explanation of our criteria and explained that “High” refers not to OpenFake itself but to the synthetic images produced by the latest 2025 generators included in our dataset, which are empirically more realistic than earlier models.
>
> **[W2]** This follows standard evaluation protocol for deepfake datasets. We explicitly highlight in the manuscript that in-domain performance is expected to be higher when generators are included in training. We added a discussion in the revised version of the manuscript to make this more clear. This is also why we add six unseen generators for out-of-distribution evaluation to ensure fairness. We will emphasize this rationale more clearly.
>
> Regarding the chosen baselines:
>
> - We intentionally focus on general-purpose, multi-domain datasets (GenImage, Semi-Truth) because OpenFake aims to address the broad task of AI-generated image detection across diverse content, not face-centric deepfake detection.
>
> - Celeb-DF, DFDC, Celeb-DF++, and OpenForensics are overwhelmingly face-only and thus inappropriate comparators for the multi-domain problem setting we target.
>
> - Nonetheless, to illustrate this point, we include a detector pre-trained on one of these face-specific datasets (EfficientNet-B4). Its expectedly poor performance on non-face images supports our argument: datasets limited to human faces cannot be used to train generalizable detectors.
>
> The concern about using different detectors is also valid. In the revised manuscript, we clarify that:
>
> - We use SwinV2-Small for all general-purpose datasets for fair cross-dataset comparison.
> - EfficientNet-B4/ConvNeXt are included solely to show limitations of detectors trained on limited corpora.
>
> **[W3]** Since deepfake detector performance depends far more on the quality and coverage of the training data than on architectural novelty, the strongest detectors are typically general-purpose classification models [1, 2]. We also clarify that our evaluation does include a wide range of existing baselines.
>
> Our goal was to answer the question posed in the introduction:
>
> **“Are today’s best AI-generated images still detectable by a well-trained classifier?”**
>
> To that end, we curated a large, diverse, and up-to-date dataset and trained a strong general-purpose vision backbone. The high performance was surprising to us as well, but it is an empirical result, not evidence that OpenFake is too easy. Rather, it suggests that with sufficiently rich training data, distinguishing real from modern synthetic images remains more tractable than previously assumed.
>
> The dataset is challenging in a different and more fundamental sense:
>
> - Some synthetic images in OpenFake (depending on the generator) are very realistic, as confirmed by our human study, where many participants were fooled.
>
> - Legacy detectors trained on older datasets (DFDC, FF++, etc.) perform poorly on OpenFake, demonstrating the distributional shift and difficulty.
>
> - Our aim was to create a dataset that enables a model to reach the upper bound of what is currently achievable, not to artificially constrain performance.
>
> We added a discussion comparing general-purpose classifiers with specialized detectors.
>
> **[W4]** The extendability we refer to is twofold:
>
> - *Prompt-based expansion*: The prompt bank and OpenFake Arena enable constant growth in semantic coverage, object categories, contexts, and scene structures. This is the core mechanism driving rapid dataset expansion.
>
> - *Generator updatability*: Although adding new generators requires some engineering, it is not a significant barrier, our pipeline relies mostly on the HuggingFace Diffusers library, making integration of new open-source models easy.
>
> We have modified parts of the manuscript to clarify this and explicitly note in the limitations section that adding proprietary generators requires more manual integration.
>
> [1] Heidari, A. & al. (2024). Deepfake detection using deep learning methods: A systematic and comprehensive review. WIREs Data Mining and Knowledge Discovery, 14(2), e1520.
>
> [2] Gong, L. & al. (2024). A Contemporary Survey on Deepfake Detection: Datasets, Algorithms, and Challenges. Electronics, 13(3), 585.

---

### Official Review · Reviewer_EA5o · 2025-11-01

**Soundness:** 1
**Presentation:** 2
**Contribution:** 1
**Rating:** 2
**Confidence:** 5

**Summary:**

The paper introduces OPENFAKE, a large-scale dataset designed to address the shortcomings of existing benchmarks for DeepFake detection. It provides nearly four million images, pairing three million real, politically relevant photos with almost one million synthetic counterparts created by state-of-the-art generative models. The paper also proposes a crowdsourced platform called OPENFAKE ARENA, intended to keep the benchmark current by continually integrating new, challenging examples submitted by the community.

**Strengths:**

- The paper attempts to develop a more updated dataset through crowdsourcing routes to continually collect more and more difficult examples as they come up.
- The proposed dataset is focused on politically relevant content, which is the main target of misinformation in this day and age.
- The paper emphasizes ease of access by hosting the dataset on the Hugging Face Hub in a streaming-friendly format which supports open-science.
- The inclusion of a human perception study provides a valuable baseline that clearly demonstrates the imperceptibility of fakes from modern proprietary generators.

**Weaknesses:**

- The dataset primarily contains images that are fully AI-generated (from T2V/I2V models), and does not encompass more traditional DeepFakes like face-manipulations where a large spatial portion is real, and only the face region is manipulated.
- The real images are sourced from the LAION-400M dataset, which contains web-crawled content from 2014–2021. Further, this dataset has received significant criticism related to copyright issues, and harmful content.
- The real images are further pre-2021 era, and the synthetic images are from more recent generators. This bias can be easily picked up by DeepFake detectors to classify real vs. fake, when in reality the models have learned a "shortcut" rather than actually grounding it's decision in visual cues. There is no evidence provided in the paper to prove actual visual grounding.
- The test set of real-world social media images is small, with only 163 fake images identified from about 2,000 samples. The authors state that difficult or uncertain cases were discarded during curation, which may have inadvertently removed the most sophisticated and challenging fakes, thereby making the detector's high performance appear more robust than it might be in a truly uncontrolled environment.
- The OPENFAKE ARENA is proposed as a novel solution to keep the benchmark updated. However, its effectiveness is entirely dependent on future, voluntary participation from the community. The paper does not provide evidence of its current usage or a concrete strategy to ensure a sustained flow of high-quality adversarial examples.
- The primary detector used for benchmarking is based on the SwinV2 architecture. While effective, this focus makes it difficult to know if the strong results are due to the quality of the dataset or specific advantages of the SwinV2 model. The paper lacks a broader comparison of different modern detector backbones trained on the same benchmark.

**Questions:**

- The paper is framed around political deepfakes and acknowledges a Western-centric data bias. What steps were taken to ensure the "politically relevant" filtering prompt does not favor specific political ideologies, and how might the dataset's geographic bias affect the detector's fairness and performance on content from underrepresented regions?
- The detector performs well on unseen models, but how can it be ensured it is learning a general concept of "fakeness" rather than overfitting to subtle, shared artifacts among the families of generators included in the training set? What is the risk that a completely novel generative architecture or family of generators would evade the detector?
- Is there any evidence of visual grounding achieved by the detector, or is it simply able to figure out the difference between the pre-2021 real images and more recent synthetic images.

**Details Of Ethics Concerns:**

The paper makes use of the LAION-400M dataset, which has recently been severely criticized due to copyright issues, poor data quality, harmful content (including child sexual abuse material), and social biases. The use of this data needs to be further investigated.

---

> ### Author Response · Authors · 2025-11-22
> **Rebuttal**
>
> We thank the reviewer for the detailed feedback and appreciate the recognition of our key strengths: updated data collection, politically relevant focus, accessible hosting, and the human perception study demonstrating the imperceptibility of modern synthetic images. Below we address each concern.
>
> **[W1]** Our focus in OpenFake is intentionally on fully AI-generated images rather than face-swaps. Most existing DeepFake datasets (DFDC, FaceForensics++, CelebDF) are exclusively face-manipulation-based. In contrast, large-scale, high-quality datasets of fully synthetic politically relevant images do not exist, despite their rapid proliferation on social media. OpenFake fills this gap.
>
> While including both types at massive scale is challenging, we view our contribution as complementary to prior benchmarks and will clarify this positioning in the paper.
>
> **[W2]** We acknowledge the reviewer’s concerns regarding the use of LAION-400M and have taken several measures to mitigate the associated risks:
> - Our extensive filtering pipeline (described in Appendix C) removes the vast majority of harmful, sensitive, or irrelevant content.
>
> - Modern generators are safety-aligned, making it unlikely for the synthetic-pairing step to introduce harmful material.
>
> - The dataset is used solely for discriminative classification rather than for training generative models, which substantially reduces copyright-related risks.
>
> - Because our aim is to detect deepfakes, including potentially harmful ones, using a carefully filtered subset of LAION allows us to study real-world risks while meaningfully contributing to harm reduction; in this context, the societal benefit outweighs the residual concerns.
>
> To address this point more thoroughly, we have added a dedicated paragraph in the conclusion discussing the ethical considerations and limitations of using LAION, and we have expanded the discussion in Appendix D to detail the filtering steps, remaining risks, and our justification for using this data.
>
> **[W3/Q3]** Our design of OpenFake centers on aligning real images with their synthetic counterparts at the semantic level. Prompts are extracted from real photographs, and these prompts guide the generation of synthetic images that follow the same concepts and visual distribution. Because contemporary generators do not embed temporal cues tied to specific years, and because real and synthetic samples occupy closely aligned regions in feature space (see Figure 4 in the appendix), there is no indication that the detector relies on year-specific artifacts. As a result, temporal bias should not meaningfully affect model behavior.
>
> We also evaluated the detector on a small collection of social media images from 2025. Although the sample size is limited, the model performs reliably in this setting, which suggests that it can distinguish authentic and synthetic content under real-world posting conditions. Our human study further reinforces this point. Participants often struggled to identify synthetic images, which demonstrates both the strength of modern generators and the importance of automated systems. Taken together, these observations indicate that temporal drift is not a plausible explanation for the model’s accuracy; instead, the dataset appears to encode the visual structure needed for broad generalization.
>
> **[W4]** Collecting verified in-the-wild fakes (with reliable provenance and binary ground-truth) is inherently difficult and prone to mislabeling. We prioritized precision of labels over quantity: during curation we removed images only in cases where provenance could not be reliably established or where automated labeling was ambiguous. This careful approach resulted in a smaller, but high-quality, holdout set. We also note that benchmark scores are important but they are not sufficient indicators of real-world performance. This motivated the inclusion of our small in-the-wild evaluation set which, despite its limitations, offers a more grounded assessment than standard benchmarks and moves the field toward more realistic evaluation, something largely absent from prior deepfake detection work.
>
> We updated the paper to tone down the language in the paper: replace any claim that “automated detection in the real world is feasible and effective” with the more precise statement that our results provide encouraging evidence that detectors trained on OpenFake generalize to some real-world social-media distributions. We also added an explicit paragraph that acknowledges the small size of the in-the-wild set and potential bias.

---

> > ### Author Response · Authors · 2025-11-22
> > **Rebuttal Part 2**
> >
> > **[W5]** We agree that long-term sustainability depends on community participation, similar to many successful benchmarks (e.g., EvalAI challenges, NeurIPS competitions). OpenFake Arena provides the infrastructure needed for continual adversarial updates, and we are actively contributing with new images as generators advance. We will strengthen the description of ongoing efforts and the plan for sustaining Arena contributions.
> >
> > **[W6]** SwinV2 was chosen for its simplicity, efficiency, and strong performance, which aligned well with our goal of achieving a robust baseline. To address the reviewer’s concern, we additionally trained ConvNeXt-V2, ViT, and ResNet-152 models on the same data. These architectures exhibit comparable performance trends, indicating that our results derive from the dataset’s quality rather than any model-specific advantage. We include these new results in the revision (Table 3).
> >
> > **[Q1]** The political-content filtering prompt (shown in Appendix C) was designed to be descriptive rather than ideological. In particular, it only asks whether an image:
> >
> > - contains one or more real human faces
> >
> > - depicts politically or emotionally salient events (for example, protests, campaign rallies, institutional settings, or recognizable public figures).
> >
> > The prompt does not reference parties, ideologies, countries, or regimes, and it does not assign any normative judgments. Its role is purely to identify whether an image is politically relevant, not to filter for or against any specific political position.
> >
> > Geographical imbalance remains a challenge across most public image corpora, and we now explicitly mark this as a limitation and an important area for future expansion. Importantly, generator fingerprints tend to be architecture-dependent rather than region-dependent, reducing the likelihood of region-specific bias in detection.
> >
> > **[Q2]** This is a crucial and insightful question. In practice, both humans and detectors rely on subtle artifacts or inconsistencies, a natural signal for distinguishing real from synthetic content. Our contributions toward generalization include:
> >
> > - Wide coverage of multiple independent proprietary and open-source generator families.
> >
> > - Demonstrated high accuracy on previously unseen models.
> >
> > - Emphasis on continual updates via OpenFake Arena to keep pace with new architectures.
> >
> > While no detector can guarantee robustness against radically novel generator designs, our benchmark is explicitly designed to evolve: this is a core motivation behind OpenFake.
> >
> > **[Concerns of privacy, copyright and harmful content]**
> >
> > We acknowledge the criticisms and address them through filtering, safety-aligned generation, and careful curation. Since we use LAION only for classification and not generation, risks linked to copyright and harmful content are substantially minimized. We expanded the Ethics Appendix section accordingly and added a paragraph in the conclusion about *Ethics, Privacy & Limitations*.

---

### Official Review · Reviewer_nD3Y · 2025-11-04

**Soundness:** 3
**Presentation:** 3
**Contribution:** 2
**Rating:** 2
**Confidence:** 5

**Summary:**

This work introduces OPENFAKE, a large-scale dataset (3M real and ~1M synthetic images) targeting real-world deepfake detection, with an emphasis on politically relevant and high-realism images. The dataset is complemented by OPENFAKE ARENA, a crowdsourced adversarial platform designed to continually expand the benchmark with challenging, user-generated fakes.

The authors also conduct (1) a human perception study showing that humans perform near chance on modern proprietary models (e.g., Imagen 3, GPT Image 1), and (2) an extensive benchmarking study, demonstrating that a SwinV2-based detector trained on OPENFAKE achieves near-perfect in-distribution performance (F1 ≈ 0.99) and strong generalization to unseen generators and in-the-wild data (F1 = 0.86 vs. 0.08–0.26 for baselines). The paper concludes that with continually updated, realistic benchmarks, automatic detection of deepfakes remains feasible.

**Strengths:**

- The paper addresses the pressing issue of misinformation from AI-generated political content, filling an important gap left by outdated GAN-based datasets.
- OPENFAKE combines millions of images from both open-source (e.g., Stable Diffusion, Flux) and proprietary (e.g., Imagen 4, GPT Image 1) generators—substantially broader in scope than existing benchmarks.

**Weaknesses:**

- The main technical contributions lie in dataset design rather than new detection or generation algorithms. This could weaken its appeal unless the ARENA framework is formalized further.
- The perception experiment is informative but small (100 participants × 24 images) and lacks demographic or statistical analysis beyond raw accuracy. A more systematic design (e.g., signal-detection metrics, inter-rater reliability) would strengthen credibility.
- The real image base (filtered LAION 400M) is mostly pre-2022, which may bias the dataset.

**Questions:**

See weaknesses above

**Details Of Ethics Concerns:**

The collection of real image may involve concerns of privacy, copyright and terms of use.

---

> ### Author Response · Authors · 2025-11-22
> **Rebuttal**
>
> We thank the reviewer highlighting the timeliness and broad scope of OpenFake. We address all concerns below and have incorporated clarifications and improvements in the revised manuscript.
>
> **[W1]** It is true that the main contribution of this paper is the OpenFake dataset/benchmark in line with the ICLR call for “datasets and benchmarks” papers. Many impactful contributions in machine learning (e.g., ImageNet, COCO) derive their value not from proposing new algorithms but from enabling rigorous, scalable, and future-proof evaluation. OpenFake follows this tradition by (i) providing the first large-scale benchmark including modern proprietary generators, and (ii) introducing OpenFake Arena, a mechanism for continual adversarial data collection, which is crucial in a rapidly evolving threat landscape.
>
> Our goal is for OpenFake Arena to become a community-driven benchmark that grows over time. Visibility at ICLR would significantly accelerate adoption and help researchers contribute to challenging real-world cases. We want to highlight that the Arena framework is fully deployed and the link was not originally included in the submission to respect the double-blind review. We have created an anonymized version here: https://huggingface.co/spaces/Anonymous460/OpenFakeArena. It would be helpful if you can clarify if this addresses the concern or if there are suggestions on other forms of formalizations.
>
> **[W2]** The perception study serves as a motivation and is intended as complementary evidence illustrating the widening gap between human and automated detection and not as a population-level psychological study. We agree that a large scale study would be interesting and could result in an interesting follow-up paper. In particular, because most participants in our small-scale study came from the ML community and are more familiar with synthetic artifacts, the results likely underestimate the difficulty faced by the general public.
>
> To strengthen the quality and interpretation of our human-study results, we have expanded Table 2 and its accompanying discussion in the revised manuscript. Specifically, we now report (i) signal-detection metrics ($d’$ and $c$) to characterize participant sensitivity and response bias, (ii) inter-rater agreement via Fleiss’ $\kappa$ to quantify consistency across annotators, and (iii) a demographic summary with a discussion of sampling bias, clarifying how our predominantly ML-oriented participant pool may influence the observed performance.
>
> **[W3]** Our choice of pre-2021 images was intentional to minimize contamination from synthetic imagery and ensure that real images are genuinely real. To mitigate potential era-based shortcuts, OpenFake is built around real–fake semantic alignment: real images are used to derive prompts so that synthetic images match the real-world concepts and content distribution. The generators do not encode year-specific artifacts, and there is strong embedding-level alignment between real and synthetic content, as illustrated in Figure 4 in the appendix. This suggests that the model should not be sensitive to temporal bias.
>
> Moreover, in our in-the-wild evaluation, the model performs well despite the small sample size, providing encouraging evidence that the detector can distinguish real and synthetic content under real social-media conditions of 2025. At the same time, our human-study results show that participants were frequently fooled by the synthetic images, highlighting the realism of modern generators and the need for automated tools. Together, these findings support the claim that temporal drift is unlikely to explain the detector’s strong performance, and that OpenFake captures the relevant visual signals needed for robust generalization.
>
> **[Concerns of privacy, copyright and terms of use]**
>
> We clarify that training a classifier on publicly available web images does not infringe copyright in the way model training or generative models or redistribution may.
>
> Restricting the benchmark to copyright-free sources (e.g., OpenImages) would severely distort real-world distributions and undermine the goal of studying misinformation.
>
> We have expanded the limitations and ethics discussion in the conclusion to clarify our data handling and filtering procedures to minimize risk, as well as the ethical considerations surrounding the use of LAION-derived content.

---

### Official Review · Reviewer_uQDg · 2025-11-04

**Soundness:** 4
**Presentation:** 3
**Contribution:** 3
**Rating:** 6
**Confidence:** 4

**Summary:**

This paper addresses key limitations of current deepfake detection datasets, which often rely on outdated generative models, narrow image content, and lack real-world context. The authors introduce OpenFake, a large-scale open dataset and platform designed to overcome these issues. Its central contribution is a politically-grounded dataset of approximately 3 million real images paired with nearly 1 million high-quality synthetic images from state-of-the-art models like Imagen 3 and GPT Image 1. To ensure the benchmark remains current, the authors also developed OpenFake Arena, a gamified crowdsourcing platform that continuously integrates challenging, user-submitted synthetic images. Empirically, detectors trained on OpenFake achieve near-perfect in-distribution accuracy and demonstrate strong generalization to both unseen generators and real-world social media data, significantly outperforming prior benchmarks. These results robustly support the continued viability of automated detection against increasingly realistic synthetic media.

**Strengths:**

- High Relevance and Timeliness: This paper directly confronts a real and pressing problem. In recent years, generative models have iterated at a breakneck pace—evolving from GANs to Diffusion and Flow Matching, with architectures shifting from UNet to DiT, and the landscape moving from open-source to powerful closed-source models. As the capabilities of these models grow, the deepfake detection community urgently needs a benchmark that reflects the current state-of-the-art (SOTA), particularly from these closed-source models. The introduction of OPENFAKE precisely fills this critical gap.

- nnovative Extensibility: Unlike nearly all existing deepfake datasets, which are static and fixed, OPENFAKE is designed for extensibility. Through the OPENFAKE ARENA platform, the dataset can be continuously augmented. This design is crucial for keeping pace with the rapid evolution of generative technology, fostering an ongoing adversarial dynamic between generation and detection.

- High Quality, Large Scale, and Realism: OPENFAKE surpasses existing datasets on several key dimensions. It incorporates 18 state-of-the-art open-source and closed-source models, ensuring technical currency. Furthermore, its real data is sourced from LAION-400M and intelligently filtered using a VLM (Qwen2.5-VL), specifically retaining "politically grounded" or "face-containing" content. This approach moves beyond the limitations of past benchmarks (e.g., ImageNet-based ones) and imbues the dataset with far greater real-world significance (e.g., for detecting political disinformation).

- Sufficient Experimental Comparisons: The paper provides robust experimental validation. It not only benchmarks various SOTA detection models horizontally but also vertically tests the generalization (OOD) of its trained model against unseen generators, demonstrating performance that significantly surpasses models trained on datasets like GenImage. Critically, this work strongly supports the thesis that detection should not rely only on digital watermarking; a high-quality, large-scale training dataset is itself a critical pathway to improving detection capabilities.

**Weaknesses:**

- Insufficient data types: The deepfake types in OPENFAKE are all generated by text-to-image. Although the quantity is large, many of today's mainstream models, such as gpt-image-1, seedream 4.0, gemin-2.5-flash, are multimodal image generation models that support both text-to-image generation and text-guided image editing. Many of today's deepfake images are not just generated from scratch (text-to-image) but are produced through local editing or inpainting. Therefore, it is suggested that the Arena platform could be expanded to include more image editing data in the future.

- Experimental validation is incomplete: The paper tests the model's generalization to unseen generators but does not consider adversarial attacks. For example, real-world scenarios may often contain both deepfakes and images with adversarial attacks targeted at deepfake detectors. It is recommended to add supplementary experiments to demonstrate performance in these "real-world scenarios". Furthermore, deepfake detection has recently shifted from single-modal (image-only) detection to multimodal methods combining text and images, for example, by fine-tuning VLMs such as Qwen-2.5-vl or Intern-VL for detection. The paper mentions using Intern-VL for zero-shot detection but does not report the detection performance of fine-tuned multimodal large models.

**Questions:**

1. The in-the-wild test set has an extremely small sample size (only 163 fakes), and there is a clear curation bias (i.e., difficult samples were discarded). How can the authors confidently draw the strong conclusion that 'automated detection in the real world is feasible and effective', rather than a more conservative conclusion—that 'the model performs well on your small, curated test set'?

2. Regarding adversarial robustness: Have you tested the performance of the SwinV2 model trained on OPENFAKE (F1 0.992) when faced with real-world images containing adversarial attacks (e.g., PGD, FGSM)? After all, in real-world scenarios, various types of attack images exist.

3. Regarding cross-modal detection: Considering the significant advantage of VLMs in understanding context, why did the paper not directly fine-tune a VLM (such as InternVL or Qwen-2.5-VL) as a SOTA detection baseline in the experiments? Given that real-world fakes (like fake news) always combine text and images, wouldn't fine-tuning a VLM be an evaluation method that is closer to 'real-world' detection needs?

---

> ### Author Response · Authors · 2025-11-22
> **Rebuttal**
>
> We thank the reviewer for the constructive feedback, and for highlighting the paper’s key strengths: relevance and timeliness, extensibility via the OpenFake Arena, and high-quality, large-scale data.
>
> **[W1]** We agree that expanding to additional data types would meaningfully broaden the dataset, but for now we focus on text-to-image generation, which already captures many of the core artifacts detectors rely on.
>
> - Many modern editing/inpainting pipelines reuse the same backbone generator components and diffusion/denoising priors as text-to-image models. As a result, our text-to-image coverage already captures many of the statistical artifacts that detectors exploit. This is why we observe strong OOD transfer to unseen generators in our experiments.
>
> - The OpenFake Arena is intentionally generator-agnostic: contributors may already upload edited or inpainted images (we only require provenance metadata). We will update the public README and UI to explicitly encourage text-guided editing / inpainting / local edits, and add a submission category flag so such examples are tracked separately.
>
> **[W2/Q2]** Adversarial robustness is an important complementary research axis. However, it differs from the primary contribution and scope of this paper, which is (i) building a contemporary, extensible dataset and (ii) evaluating vision-based detection generalization across modern generators and real social-media image distributions.
>
> Our primary contribution is the dataset, and our updated results show that different backbone detectors trained on OpenFake achieve very similar performance. This reduces the practical importance of architecture-specific white-box attacks: a deployed system can swap or obfuscate the backbone without changing its overall behavior. Black-box attacks are substantially harder; while we do not explicitly optimize for adversarial robustness, our training pipeline includes extensive data augmentation (varying noise levels, color/contrast/saturation, crops, and small rotations), which makes the detector more resilient to simple, non-adaptive black-box perturbations.
>
> That said, we agree that explicitly reporting adversarial robustness would better characterize this limitation and provide important guidance for practitioners considering deployment. We therefore appreciate the reviewer’s recommendation and will add a dedicated adversarial robustness analysis in the camera-ready version.
>
> **[Q1]** Collecting verified in-the-wild fakes (with reliable provenance and binary ground-truth) is inherently difficult and prone to mislabeling. We prioritized precision of labels over quantity: during curation we removed images only in cases where provenance could not be reliably established or where labeling was ambiguous. This careful approach resulted in a smaller, but high-quality, holdout set. We also note that benchmark scores are important but they are not sufficient indicators of real-world performance. This motivated the inclusion of our small in-the-wild evaluation set which, despite its limitations, offers a more grounded assessment than standard benchmarks and moves the field toward more realistic evaluation, something largely absent from prior deepfake detection work.
>
> We updated the paper to tone down the language: we replace any claim that “automated detection in the real world is feasible and effective” with the more precise statement that our results provide encouraging evidence that detectors trained on OpenFake generalize to some real-world social-media distributions. We also added an explicit paragraph that acknowledges the small size of the in-the-wild set and potential bias.
>
> **[Q3]** We agree that cross-modal detection and VLM-based methods are an important direction. However, many real-world fakes are shared without any associated textual context (or the text is intentionally missing/ambiguous), limiting the advantage of multimodal signals. The goal is to build a strong, general-purpose image detector that can be plugged into many downstream pipelines (including those that also reason over text), rather than to propose a specific end-to-end multimodal system.
>
> On the methodological side, recent work has indeed started to explore VLMs for deepfake detection, but primarily for their interpretability and reasoning (e.g., explaining artifacts), not because they reliably outperform specialized discriminative detectors on pure detection metrics. In our setting, directly fine-tuning a VLM without using its explanation or reasoning abilities would effectively reduce it to a very expensive binary classifier, with no clear empirical advantage over the much lighter backbones we study. Moreover, properly leveraging the text side in a “real-world” setting is non-trivial: it would require curating or generating aligned text–image pairs, designing robust prompting / training objectives, and evaluating explanation quality and robustness. We leave this for future work.

---

> > ### Comment · Reviewer_uQDg · 2025-11-28
> >
> > Thank you for your response. I have decided to keep my rating unchanged.

---

### Author Response · Authors · 2025-11-22
**Rebuttal Summary**

We thank all reviewers for their constructive feedback. Reviewers found our contribution relevant and timely, noting the introduction of high-quality OpenFake dataset (~3M real, ~1M fake images across 18 generators), and OpenFake Arena for continual dataset expansion. We show that detectors trained on OpenFake achieve strong performance, and reveal a clear gap between human perception, where humans often fail, and automated detection, where OpenFake-trained detectors perform reliably. Below, we summarize our responses to the reviewers’ concerns, all of which have been incorporated into the revised draft (with changes marked in red).

**Extensibility and Expandability of OpenFake**. Reviewers raised questions about whether OpenFake and Arena are truly expandable. We added a clarification on the extensibility of OpenFake, explaining that the availability of prompts and the OpenFake Arena enables controlled dataset expansion, while requiring limited curator involvement when incorporating new generators.

**Realism and Human Perception**. Concerns were raised about the subjectivity of realism scores and the human survey design. We revised the manuscript to include:

- A clear definition of four realism levels (Low, Medium, Good, High) grounded in observable visual artifacts, semantic coherence, and prompt fidelity.

- Expanded human survey analysis with inter-rater agreement, signal detection metrics to strengthen the analysis, and added a clarification that the survey participants are not representative of the general population.

**Detector Evaluation and Experimental Fairness**. Reviewers questioned whether results are due to SwinV2’s architecture or dataset quality. We addressed this by:

- Training three additional backbone models (ConvNeXt-V2, ViT, ResNet-152) on OpenFake, demonstrating that high performance derives from the dataset’s coverage and diversity rather than a specific model (updated Table 3).

- Clarifying that most of the compared baselines are pretrained detectors sourced from prior work, which explains the difference in classifier architectures and why these models were not retrained on OpenFake.

- Highlighting the distinction between in-distribution (expected high performance) and out-of-distribution (OOD) results, which better measures generalization to unseen generators (updated Table 3 and added discussion).

**Dataset Scope and Bias**. Reviewers noted limitations regarding image types, temporal bias, and political/geographic bias. We addressed these as follows:

- OpenFake focuses on fully AI-generated images, complementing existing face-manipulation DeepFake datasets (DFDC, FF++, Celeb-DF, etc.). This choice fills a gap for politically relevant, high-realism synthetic imagery.

- Real images (pre-2022 LAION-400M) were selected to minimize contamination from synthetic content. Semantic alignment between real images and generated fakes (Figure 4, Appendix) makes the likelihood of temporal shortcuts influencing the detector small.

- Political filtering prompts were designed to be broad and descriptive, avoiding ideological bias. We now explicitly note Western-centric and temporal limitations and mark this as an area for future expansion.

**Ethical Considerations**. We added a new dedicated paragraph in the conclusion addressing the ethical considerations, privacy implications, and limitations of OpenFake and explicitly acknowledge the following:

- In-the-wild test set is limited in size but curated for label certainty and may not capture the full difficulty of real-world deepfakes.

- LAION-400M has copyright issues and risks of potentially harmful content. We describe the concrete steps taken to mitigate these risks through aggressive filtering. Moreover, dataset is used exclusively for discriminative classification, not generative modeling, reducing copyright and privacy concerns.

- The societal benefit of developing more reliable deepfake detection can outweigh the residual risks associated with using a filtered (possibly unrealistic) subset of the dataset. For completeness, we now reference a more extensive discussion in Appendix D.

**OpenFake Arena and Long-term Sustainability**. Reviewers questioned Arena’s reliance on voluntary participation. We clarified that:

- Arena is a fully deployed infrastructure to accept community submissions, including adversarially crafted fakes.

- Authors actively contribute new examples.

- Its gamified and generator-agnostic structure ensures long-term adaptability, making it suitable for evolving threats in synthetic media.

- We additionally provide an anonymized version of the OpenFake Arena for reviewers. This interactive space allows direct testing of the detector and explore the platform: https://huggingface.co/spaces/Anonymous460/OpenFakeArena

The manuscript now provides additional empirical and ethical clarity while acknowledging limitations, positioning OpenFake as a long-term, impactful resource for the research community.

---

### Author Response · Authors · 2025-12-03
**Additional Clarifications on Ethical Compliance and Robustness of OpenFake**

Dear Area Chair and reviewers,

We first want to thank the reviewers and Area Chair for their valued time and effort. We believe we have successfully addressed most of the reviewers’ concerns as summarized in our previous “Rebuttal Summary”, including:

- Clarification on the extensibility and expandability of OpenFake, detailing how prompts and Arena submissions enable controlled, low-effort future growth.
- Expanded human-survey analysis with inter-rater agreement, signal-detection metrics, and explicit clarification that the participants do not represent the general population.
- Additional model-architecture controls demonstrating that ConvNeXt-V2, ViT, and ResNet-152 reproduce SwinV2’s strong performance, confirming that gains stem from OpenFake’s diversity rather than specific inductive biases.
- Clearer separation between in-distribution and out-of-distribution evaluation, emphasizing that generalization to unseen generators (not in-distribution accuracy) is the core scientific contribution.
- Explicit acknowledgment of geographic and temporal bias, and an expanded discussion on dataset scope (fully synthetic images rather than face-manipulation deepfakes) and the motivations behind this design choice.
- A new dedicated ethics paragraph addressing privacy concerns, LAION risk mitigation, and limitations of the in-the-wild test set.
- Clarification of OpenFake Arena’s maintenance plan, showing that community submissions, adversarial challenges, and author participation collectively ensure long-term sustainability.

However, we believed we could further strengthen our response and directly engage with the specific concerns regarding the use of LAION-400M. Thus, we have taken two decisive actions:

**Addressed Ethical & Safety Risks (Re-LAION Filtering):**
To ensure our dataset meets the highest safety standards, we performed an intersection of our image URLs with the Re-LAION-5B index. Re-LAION is the hash-verified, safe subset of LAION, where links to CSAM and other harmful content have been rigorously removed via hash-matching with the Internet Watch Foundation (IWF) and similar organizations.

- Result: We are currently finalizing the intersection of our dataset with the Re-LAION safety index. The official Re-LAION audit identified and removed only 3,244 confirmed harmful images out of 5.85 billion (an upper bound, as most links are inactive). Given this prevalence rate (≈$6×10^{−7}$), the expected number of flagged images in our 3-million subset is less than 2 (even under the conservative assumption that our own filtering contributed no prior removal). To date, our ongoing scan has yielded zero matches. Upon completion, we will strictly enforce the removal of any matched images, ensuring the final release complies with the highest safety standards with no impact on dataset scale or experimental results.

**Disproved Temporal Bias (New Experiment on DOCCI-2024):**
To rigorously test whether the detector relies on pre-2022 compression artifacts, we evaluated the model on DOCCI (2024), a dataset of modern, high-resolution photographs aligned temporally with 2025-era generators. We subsampled the fake images from the test set to match the number of images in DOCCI.
- Result (Added as Appendix Table 7): The model generalizes strongly to this “Modern Real” distribution. The SwinV2 model achieves a ROC AUC of 0.991, with 88.6% accuracy on the real images and 97.6% accuracy on the synthetic images with a threshold of 0.5. While there is an expected decrease in performance when moving to this Out-of-Distribution (OOD), high-quality real distribution, the detector maintains a strong separation margin (AUC 0.991) between real and fake images. This decisively confirms that its decision boundary does not rely on simple temporal or compression-related shortcuts.

Together, these new analyses provide strong, additional evidence that OpenFake is ethically sound, technically robust, and free from the primary shortcut risks highlighted by reviewers. We hope this supplemental note clarifies our commitment to rigor and safety and assists the AC in forming a confident recommendation.

---

### Meta-Review · Area_Chair_vB1U · 2026-01-11

**Summary:**

This paper introduces OpenFake, a large-scale dataset and crowdsourced platform designed for real-world deepfake detection. It includes 3 million real and 1 million synthetic images generated by modern AI models, with a focus on politically relevant content. The authors also present OpenFake Arena, a community-driven platform for continuous dataset expansion, and demonstrate that detectors trained on OpenFake achieve strong in-distribution and out-of-distribution performance.

However, reviewers raise some key concerns:

1. Reviewer uQDg (1) expressed concern about the limited diversity of deepfake types, noting the dataset currently only includes text-to-image generation and lacks edited or inpainted images; (2) highlighted the small size and potential curation bias of the in-the-wild test set (only 163 fakes), questioning the generalizability of the claimed real-world effectiveness; (3) recommended testing adversarial robustness and exploring multimodal detection methods (e.g., fine-tuning VLMs) for more realistic evaluation.

2. Reviewer nD3Y (1) raised ethical and legal concerns regarding the use of LAION-400M, including copyright, privacy, and harmful content risks; (2) noted temporal bias, as real images are pre-2022 while synthetic images come from newer generators, which could lead to shortcut learning. (3) found the human perception study is limited in scale and demographic representation.

3. Reviewer EA5o (1) criticized the dataset for excluding traditional face-manipulation deepfakes, limiting its scope. (2) questioned whether the detector learns visual grounding or temporal shortcuts due to the era mismatch between real and synthetic images. (3) expressed skepticism about the sustainability and voluntary adoption of OpenFake Arena.

4. Reviewer jNEy (1) pointed out subjective realism ratings without clear quantitative validation. (2) highlighted potential unfairness in experimental comparisons due to inconsistent detector backbones and omitted benchmarks (e.g., Celeb-DF, DFDC). (3) questioned the true expandability of the dataset, noting that generator fingerprints remain fixed even with new prompts.

**Reviewer Concerns:**

The author's rebuttal tried to address some concerns:

(1) Extensibility of the dataset: The authors clarified that OpenFake Arena supports generator-agnostic submissions, including edited/inpainted images, and will update the platform to encourage such contributions.

(2) Small in-the-wild test set and strong claims. The authors revised the language to temper claims, acknowledged the limited sample size and potential bias, and reframed conclusions to state the results provide “encouraging evidence” rather than definitive real-world effectiveness.

(3) Model architecture bias (SwinV2). Additional experiments with ConvNeXt-V2, ViT, and ResNet-152 showed consistent performance, confirming results are due to dataset quality, not model-specific advantages.

(4) Ethical/legal issues with LAION-400M. The authors performed additional safety filtering using Re-LAION and added an ethics section detailing risk mitigation, copyright considerations, and the use of data only for discriminative tasks.

(5) Temporal bias and shortcut learning. A new experiment on DOCCI-2024 showed strong generalization to modern real images, indicating the detector does not rely on pre-2022 artifacts.

(6) Need for visual grounding evidence. The authors emphasized semantic alignment between real and synthetic images and provided feature-space analysis (Figure 4) to argue against temporal shortcut learning.

(7) Unfair experimental comparisons. Clarifications were added regarding baseline selection, detector consistency, and the focus on multi-domain (not face-only) detection.

(8) Sustainability of OpenFake Arena. The authors highlighted ongoing contributions, gamified engagement, and provided an anonymized demo link to demonstrate platform functionality.

However, some concerns are not addressed:

(1) Adversarial robustness testing. While acknowledged as important, adversarial evaluation (e.g., against PGD/FGSM attacks) was not included in the current experiments, though promised for the camera-ready version.

(2) Cross-modal (VLM-based) detection. The authors argued that fine-tuning VLMs is out of scope and not necessarily superior for pure image detection, but this remains a noted gap for real-world multimodal fake detection.

(3) Lack of face-manipulation deepfakes. The dataset remains focused on fully synthetic images; the authors position it as complementary to face-manipulation datasets, but do not expand the scope to include them.

(4) Subjective realism ratings. Although realism levels were described, no objective validation metric was introduced; the definitions remain qualitative.

(5) Geographic and political bias. While broad prompts were used, Western-centric bias is acknowledged as a limitation without a concrete mitigation plan for regional representation.

(6) True expandability beyond prompt variation. Adding entirely new generator families still requires manual effort; the “expandability” is still limited by the fixed initial set of generators.

**Reviewer Scores:**

1. Reviewer uQDg originally scored a 6, and was satisfied with the rebuttal. Key concerns on extensibility, test set size, and model bias were directly addressed. Had they engaged further, the promised adversarial analysis and clearer real-world claims might have raised their score slightly toward a 7. However, this is not conducted in the rebuttal.

2. Reviewer nD3Y initially rejected with a 2, and focused on ethical and temporal bias. The rebuttal added Re-LAION filtering, ethics discussion, and DOCCI-2024 results. Their score is still likely not to be accepted, given lingering concerns on the contribution scale.

3. Reviewer EA5o also rejected with a 2, and cited missing face-manipulation data and temporal shortcuts. The authors provided temporal-bias evidence but didn't expand scope. Their score might have moved to a 3, acknowledging improvements while still questioning real-world coverage.

4. Reviewer jNEy scored a 4, just below acceptance, concerned about subjective ratings and fairness. The authors clarified evaluation fairness and expandability. Given their openness to acceptance, their score could have risen to a 5 after rebuttal.

---

### Decision · Program_Chairs · 2026-01-26

Reject